# Fragment and Geometry Aware Tokenization of Molecules for Structure-Based Drug Design Using Language Models

**Cong Fu**[1][*]**, Xiner Li**[1][*]**, Blake Olson**[1]**, Heng Ji**[2]**, Shuiwang Ji**[1]
[1] Texas A&M University, College Station, TX, USA
[2] University of Illinois Urbana-Champaign, Champaign, IL, USA
{congfu, lxe, blakeolson, sji}@tamu.edu
{hengji}@illinois.edu

## Abstract

Structure-based drug design (SBDD) is crucial for developing specific and effective therapeutics against protein targets but remains challenging due to complex protein-ligand interactions and vast chemical space. Although language models (LMs) have excelled in natural language processing, their application in SBDD is underexplored. To bridge this gap, we introduce a method, known as Frag2Seq, to apply LMs to SBDD by generating molecules in a fragment-based manner in which fragments correspond to functional modules. We transform 3D molecules into fragment-informed sequences using $SE(3)$-equivariant molecule and fragment local frames, extracting $SE(3)$-invariant sequences that preserve geometric information of 3D fragments. Furthermore, we incorporate protein pocket embeddings obtained from a pre-trained inverse folding model into the LMs via cross-attention to capture protein-ligand interaction, enabling effective target-aware molecule generation. Benefiting from employing LMs with fragment-based generation and effective protein context encoding, our model achieves the best performance on binding vina score and chemical properties such as QED and Lipinski, which shows our model's efficacy in generating drug-like ligands with higher binding affinity against target proteins. Moreover, our method also exhibits higher sampling efficiency compared to atom-based autoregressive and diffusion baselines with at most $\sim 300\times$ speedup. The code will be made publicly available at `https://github.com/divelab/AIRS/tree/main/OpenMI/Frag2Seq`.

## 1 Introduction

Structure-based drug design (SBDD) is a critical method in medicinal chemistry that involves the design and optimization of molecules to interact specifically and effectively with biological targets, typically protein pockets (Anderson, 2003). This approach is fundamental in developing new therapeutic drugs as it allows for more precise interaction with biological systems, potentially reducing side effects and increasing efficacy. Traditionally, SBDD relies heavily on high-throughput virtual screening (Lyne, 2002; Shoichet, 2004) and experimental validation. These processes are not only time-consuming and labor-intensive but also require substantial financial resources.

In recent times, machine learning has emerged as a promising approach for advancing scientific endeavors (Zhang et al., 2023b; Liu et al., 2021b), such as physical simulation (Sanchez-Gonzalez et al., 2020; Helwig et al., 2023; Fu et al., 2024a; Zhang et al., 2024b; Gui et al., 2025), quantum mechanics (Kochkov et al., 2021; Fu et al., 2022), materials (Yan et al., 2024), and biology (Liu et al., 2021a; 2022a;b; Wang et al., 2022; Fu et al., 2024b). SBDD poses great challenges to machine learning models as it requires the model to capture complicated protein-ligand interaction while improving the drug-likeness of designed molecules. Earlier works have tried to use autoregressive models (Luo et al., 2021; Peng et al., 2022) and diffusion models (Guan et al., 2022; 2023; Huang et al., 2024b; Li et al., 2024b; Zhou et al., 2024) to encode context information and generate molecules.

---

[*]Equal contribution

However, these models only consider atom-wise generation, and diffusion models typically need thousands of steps to generate, which results in an inefficient generation. Another line of work chooses to generate molecules based on molecular fragments (Zhang et al., 2022; Zhang & Liu, 2023), but it often requires a complicated pipeline involving several neural networks to choose and link fragments.

Recently, language models (LMs) have demonstrated substantial promise in various fields due to their robust data processing and generative capabilities (Vaswani et al., 2017; Devlin et al., 2018; Brown et al., 2020; Gu et al., 2021; Wang et al., 2024; Khurana et al., 2024). These models, especially large language models (LLMs) (Zhang et al., 2024a; Touvron et al., 2023; Achiam et al., 2023; Chowdhery et al., 2023; Huang et al., 2024a), excel in learning complex patterns and producing coherent outputs, which makes them ideal candidates for advanced tasks in natural language processing and beyond. Despite these strengths, the application of LMs to SBDD remains largely unexplored. LMs offer notable advantages, such as handling large datasets with prominent efficiency over diffusion-based methods, learning from massive biological and chemical texts for diverse potential tasks, and generating molecules and materials (Li et al., 2024a; Yan et al., 2025). However, challenges persist, particularly the need to adapt LMs to process geometric graph structures of molecular data, which are fundamentally different from textual data. Moreover, it's promising to consider how to apply LMs to perform fragment-based generation, which can generate more realistic substructures and reduce generation steps to improve efficiency. Additionally, ensuring that these models can accurately simulate the physical and chemical properties of molecules remains a significant hurdle, and it's also crucial to consider how to encode protein context information in LMs to effectively capture protein-ligand interaction in order to generate molecules that can bind to target proteins. Addressing these challenges could unlock the transformative potential of LMs in SBDD, leading to more efficient and effective drug discovery processes.

In this work, we propose to employ LMs to generate molecules in a fragment-based manner for the SBDD task. To achieve this, we develop a novel approach to convert 3D molecules into fragment-informed sequences by constructing $SE(3)$-equivariant molecule and fragment local frames and then extracting $SE(3)$-invariant sequences that contain the $SE(3)$-invariant coordinates and orientations of 3D fragments. To consider protein conditions, we use an existing pre-trained inverse folding model to extract the embedding of protein pockets and incorporate this information into LMs by cross-attention mechanism. In this way, our method enables the usage of powerful LMs in target-aware molecule generation while keeping geometric information in sequence conversion. Our experimental results evidence these advantages, outperforming strong baselines in both efficacy and efficiency.

## 2 RELATED WORKS

**Structure-Based Drug Design.** Structure-based drug design aims to generate molecules that can bind to the given protein pocket. LiGAN (Ragoza et al., 2022) uses an atomic density grid to represent protein-ligand structures. The atom type, positions, and bonds are constructed from generated density grids. Then, the following works (Luo et al., 2021; Peng et al., 2022; Liu et al., 2022a) generate molecules using autoregressive models. For example, GraphBP (Liu et al., 2022a) uses normalizing flow to generate each atom's relative position in an equivariant way by constructing local coordinate systems. Pocket2Mol (Peng et al., 2022) also considers bond generation to improve the structure validity. Compared with autoregressive models, diffusion-based SBDD models (Schneuing et al., 2022; Guan et al., 2022; 2023) can generate all atoms of molecules in a one-shot way. For example, DiffSBDD (Schneuing et al., 2022) denoises all atom positions and types from a Gaussian distribution. IPDiff (Huang et al., 2024b) uses a pretrained binding affinity prediction model to guide the diffusion generation. To generate more realistic substructures, FLAG (Zhang et al., 2022) and DrugGPS (Zhang & Liu, 2023) adopt fragment-based methods to generate molecules motif-by-motif. For those methods using language models, TamGen (Xia et al., 2024) only generates SMILES, and Lingo3DMol (Feng et al., 2024) generates fragment-based SMILES then predicts molecule coordinates. They both need pre-training on millions of SMILES or 3D molecule structures to achieve good results.

**Language Models for Chemistry.** Drawing inspiration from the success of LMs in NLP and beyond, chemical language models (CLMs) emerge as a competent way for representing molecules (Bran & Schwaller, 2023; Janakarajan et al., 2023; Bajorath, 2024; Zhang et al., 2024a). Variants of LMs have been adapted for molecular science, producing a variety of works including DrugGPT (Li et al.,

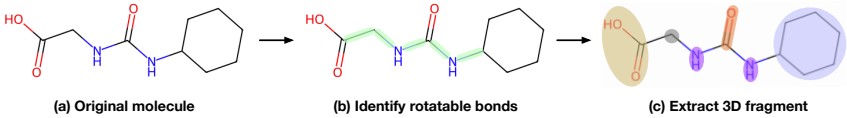

Figure 1: Illustration of splitting molecules into fragments.

2023c), DrugChat (Liang et al., 2023), MoleculeSTM (Liu et al., 2023a), ChemGPT (Frey et al., 2023) MolGPT (Bagal et al., 2021), MolReGPT (Li et al., 2023a), MolT5 (Edwards et al., 2022), MoleculeGPT (Zhang et al., 2023a), InstructMol (Cao et al., 2023), and many others (Luo et al., 2022; Mao et al., 2023b; Haroon et al., 2023; Blanchard et al., 2023). CLMs learn the chemical vocabulary and syntax used to represent molecules. All inputs including chemical structures and property syntax should be converted into a sequence form and tokenized for compatibility with language models. Commonly, SMILES (Weininger, 1988) is used for this sequential representation, although other formats like SELFIES (Krenn et al., 2019), atom type strings, and custom strings with positional or property values are also viable options. To learn representations, CLMs are usually pre-trained on extensive molecular sequences through self-supervised learning. Subsequently, models are fine-tuned on more focused datasets with desired properties, such as activity against a target protein. Most existing CLM works consider chemical structures as well as other modalities such as natural language captions (Bagal et al., 2021; Li et al., 2023a;c; Edwards et al., 2022; Xie et al., 2023; Chen et al., 2023b; Tysinger et al., 2023; Xu et al., 2023; Chen et al., 2023a; Pei et al., 2023; Liu et al., 2023b; Wang et al., 2023), while some focus on pure text of chemical literature (Luo et al., 2022) or molecule strings (Haroon et al., 2023; Mao et al., 2023b; Blanchard et al., 2023; Mazuz et al., 2023; Fang et al., 2023; Kyro et al., 2023; Izdebski et al., 2023; Yoshikai et al., 2023; Wu et al., 2023; Mao et al., 2023a). Notably, all these works solely consider 2D molecules for representation learning and downstream tasks, overlooking 3D geometric structures which is crucial in many chemical predictive and generative tasks. In order to use pivotal 3D information, another line of work incorporate geometric models such as GNNs in parallel with the CLM (Xia et al., 2023; Zhang et al., 2023a; Cao et al., 2023; Liang et al., 2023; Liu et al., 2023a; Li et al., 2023b; Frey et al., 2023), which requires additional design and training techniques to mitigate alignment issues. However, no existing work uses LMs to directly generate 3D ligands in structure-based drug design.

## 3 METHODS

In this section, we describe how to convert 3D molecules into sequences in a fragment-based manner, which can be effectively processed by language models. In Section 3.1, we formally introduce the problem definition of SBDD. Then, in Section 3.2, we show how to split 3D molecules into 3D fragments. With these 3D fragments, next in Section 3.3, we introduce the way to construct a bijective mapping between 3D fragments and $SE(3)$-invariant sequences by constructing $SE(3)$-equivariant molecule and fragment local frames. Finally, in Section 3.4, we describe how to incorporate conditional information of proteins into the language model and the training and generation strategies.

### 3.1 PROBLEM DEFINITION

Our objective is to design 3D molecules (*i.e.*, ligands) that can effectively bind to a given protein target within its binding pocket while also demonstrating suitable drug-like properties. The 3D geometry of a protein binding pocket is represented as $\mathcal{P} = \{(\boldsymbol{s}_i, \boldsymbol{b}_i)\}_{i=1}^n$, where $\boldsymbol{s}_i \in \mathbb{R}^3$ denotes the 3D Cartesian coordinates of the $i$-th atom and $\boldsymbol{b}_i$ is a one-hot vector that denotes the atom type. Similarly, we represent a ligand as $\mathcal{M} = \{(\boldsymbol{v}_j, \boldsymbol{z}_j)\}_{i=1}^m$, where $\boldsymbol{v}_j \in \mathbb{R}^3$ and $\boldsymbol{z}_j$ denote coordinates and atom type, respectively. Our goal is to learn a conditional generative model that captures the conditional probability $p(\mathcal{M}|\mathcal{P})$ from the training protein-ligand pairs.

### 3.2 3D MOLECULE FRAGMENTATION

Since we generate molecules in a 3D fragment-based manner, we first need to decompose molecules in the training set into 3D fragments. Specifically, as shown in Figure 1, we split fragments by cutting rotatable chemical bonds that meet three conditions: (1) the bond is not in a ring, (2) the bond type is single, and (3) the degree of the beginning and end atom on the bond is larger than 1. The third condition prevents breaking the functional groups, such as the hydroxy and carboxyl

groups. To maintain a canonical order of splited fragments, we further sort fragments based on the order of their appearance in the canonical SMILES representation. More details of canonical order are described in Section 3.3.1. Formally, for a molecule $\mathcal{M}$, we split it into a set of 3D fragments $\{\mathcal{G}_i = (Z_i, V_i)\}_{i=1}^{k}$, where $Z_i$ denotes atom type matrix and $V_i$ denotes Cartesian coordinates.

## 3.3 FRAGMENT-BASED 3D MOLECULE TOKENIZATION

### 3.3.1 ATOM ORDERING BASED ON 3D GRAPH ISOMORPHISM

As the first step in molecule tokenization, we need to transform a molecular (fragment) graph into a 1D sequential representation. Thus we need to establish an order for the atoms of given 3D graphs. To acheive dimension reduction with least information loss, we seek uniqueness in the ordering and resort to canonical SMILES (O'Boyle, 2012; Weininger et al., 1989)as a solution. A key property of canonical SMILES is its ability to provide a unique string representation for a given molecular structure, which is not inherently guaranteed by the basic SMILES algorithm. We refer to a set of (atom) orders with canonical properties as canonical orders (McKay et al., 1981; McKay & Piperno, 2014). The canonicalization of SMILES involves a deterministic process where the algorithm selects a unique starting atom and follows a set of predefined rules to traverse the molecule in a systematic way. This results in a consistent and reproducible ordering of atoms and bonds within the SMILES string, irrespective of the initial input format. The canonical form is crucial for database searches and for ensuring consistency in chemical databases, as it prevents duplicates and allows for efficient indexing and retrieval of molecular data, which ensures maximum invariance for atom reordering.

To formally study which molecules the canonical forms can distinguish between and not, we first give the below definition following graph theories.

**Definition 3.1.** *[3D Molecular Graph Isomorphism] Let $\mathcal{M}_1 = (V_1, Z_1)$ and $\mathcal{M}_2 = (V_2, Z_2)$ be two 3D molecular graphs, where $z_i$ is the node type vector and $v_i$ is the node coordinates of the molecule $\mathcal{M}_i$. Let $ver(\cdot)$ denote the set of vertices, $attr(\cdot)$ denote node attributes, and no edge exists. Let $\mathcal{M}_1 \cong \mathcal{M}_2$ denote two attributed graphs are isomorphic. Two 3D molecules $\mathcal{M}_1$ and $\mathcal{M}_2$ are **3D isomorphic**, denoted as $\mathcal{M}_1 \cong_{3D} \mathcal{M}_2$, if there exists a bijection $b : ver(\mathcal{M}_1) \rightarrow ver(\mathcal{M}_2)$ such that for every atom in $\mathcal{M}_1$ indexed $i$, $z_i^{\mathcal{M}_1} = z_{b(i)}^{\mathcal{M}_2}$, and there exists a 3D transformation $\tau \in SE(3)$ such that $v_i^{\mathcal{M}_1} = \tau(v_{b(i)}^{\mathcal{M}_2})$. If a small error $\epsilon$ is allowed such that $|v_i^{\mathcal{M}_1} - \tau(v_{b(i)}^{\mathcal{M}_2})| \leq \epsilon$, we call the two 3D graphs $\epsilon$-constrained 3D isomorphic.*

This leads us to the following guarantee.

**Lemma 3.2.** *[Canonical Ordering for 3D Molecular Graph Isomorphism] Let $\mathcal{M}_1 = (V_1, Z_1)$ and $\mathcal{M}_2 = (V_2, Z_2)$ be two 3D molecular graphs following Def. 3.1. Let $\boldsymbol{L} : \mathbb{M} \rightarrow \mathcal{L}$ be a function that maps a molecule $\mathcal{M} \in \mathbb{M}$, the set of all finite 3D molecular graphs, to its canonical order $\boldsymbol{L}(\mathcal{M}) \in \mathcal{L}$, the set of all possible canonical orders, as produced by the canonical SMILES. Then the following equivalence holds:*

$$\boldsymbol{L}(\mathcal{M}_1) = \boldsymbol{L}(\mathcal{M}_2) \Leftrightarrow \mathcal{M}_1 \cong_{3D} \mathcal{M}_2$$

*where $\mathcal{M}_1 \cong_{3D} \mathcal{M}_2$ denotes that $\mathcal{M}_1$ and $\mathcal{M}_2$ are 3D isomorphic.*

Lemma 3.2 indicates that the established ordering with canonical SMILES is both complete (sufficient to distinguish non-3D-isomorphic molecules) and sound (not distinguishing actually 3D-isomorphic molecules). Detailed proof is provided in Appendix C.1.

### 3.3.2 $SE(3)$-EQUIVARIANT MOLECULE AND FRAGMENT FRAMES CONSTRUCTION

To integrate 3D structure information into our sequences, one main challenge is to ensure the $SE(3)$-invariance property. Specifically, given a 3D molecule, if it is rotated or translated in the 3D space, its 3D representation should be unchanged. In order to incorporate invariance, we first need to construct $SE(3)$-equivariant frames for fragments and whole molecules.

Given a 3D molecule $\mathcal{M}$ with atom types $Z$ and atom coordinates $V$, we first build a **molecule local coordinate frame** $\mathrm{m} = (\boldsymbol{x}, \boldsymbol{y}, \boldsymbol{z})$ based on the input molecule. For the fragments $\{\mathcal{G}_i\}_{i=1}^{k}$ of $\mathcal{M}$, we sort the $k$ fragments based on the canonical order $\boldsymbol{L}(\mathcal{M})$, specifically by the order of a fragment's first-ranked atom. We calculate the average atom coordinates as the center of each fragment. As

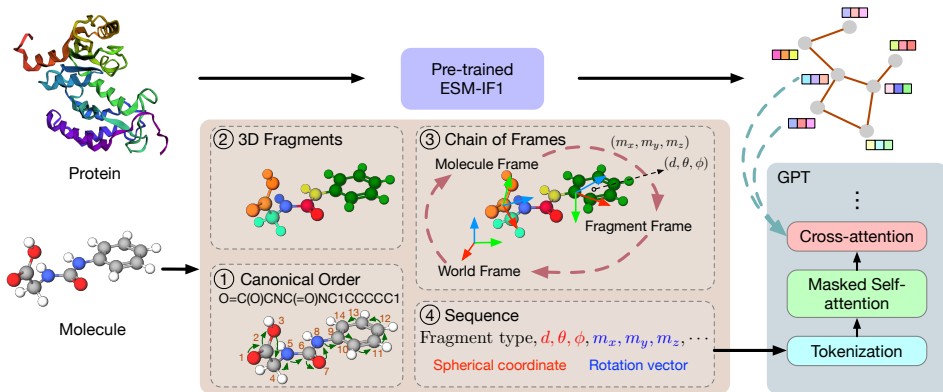

Figure 2: Overview of Frag2Seq pipeline. Atoms in the small molecule are reordered according to the order in canonical SMILES. Then, we split the 3D molecules into 3D fragments and sort them based on the canonical order. Next, we construct $SE(3)$-equivariant molecule and fragment frames, which are local coordinate systems for molecules and fragments, respectively. Then, we obtain the $SE(3)$-invariant spherical coordinates $(d, \theta, \phi)$ of each fragment center under the molecule frame and $SE(3)$-invariant rotation vector $(m_x, m_y, m_z)$ between fragment and molecule frames. Lastly, we concatenate them into sequences. Protein node embeddings are obtained using the ESM-IF1 model and incorporated into the language model via cross-attention.

shown in Figure 2, the frame is built based on the first three non-collinear fragment centers in the canonical order. Let $\ell_1, \ell_2$, and $\ell_m$ be the indices of these three fragment centers. Then the molecule local frame $\mathfrak{m} = (\boldsymbol{x}, \boldsymbol{y}, \boldsymbol{z})$ is calculated as

$$\boldsymbol{x} = \text{normalize}(\boldsymbol{v}_{\ell_2} - \boldsymbol{v}_{\ell_1}), \quad \boldsymbol{y} = \text{normalize}\left((\boldsymbol{v}_{\ell_m} - \boldsymbol{v}_{\ell_1}) \times \boldsymbol{x}\right), \quad \boldsymbol{z} = \boldsymbol{x} \times \boldsymbol{y}, \quad (1)$$

where normalize($\cdot$) is the function to normalize a vector to unit length. Note that the molecule local frame is equivariant to the rotation and translation of the input molecule.

Given the basis vectors $(\boldsymbol{x}, \boldsymbol{y}, \boldsymbol{z})$ of the molecule local frame, we can construct the transformation between the molecule local frame $\mathfrak{m}$ and the world frame $\mathfrak{w}$, which we denote as $\mathfrak{m} \to \mathfrak{w}$. Specifically, the transformation involves a rotation matrix $R_{\mathfrak{m} \to \mathfrak{w}} \in \mathbb{R}^{3 \times 3}$ with $|R_{\mathfrak{m} \to \mathfrak{w}}| = 1$ and a translation vector $\boldsymbol{t}_{\mathfrak{m} \to \mathfrak{w}} \in \mathbb{R}^3$. Since the basis vectors $(\boldsymbol{x}, \boldsymbol{y}, \boldsymbol{z})$ are already normalized and orthogonal to each other, we can directly stack them together to form the rotation matrix, such that $R_{\mathfrak{m} \to \mathfrak{w}} = [\boldsymbol{x}^T, \boldsymbol{y}^T, \boldsymbol{z}^T]$. For the translation vector, we can set $\boldsymbol{t}_{\mathfrak{m} \to \mathfrak{w}} = \boldsymbol{v}_{\ell_1}$ as it represents the displacement between the origins of the two frames.

Next, we need to build the local coordinate system for each 3D fragment. Similar to the building process of the molecule local frame, we use the first three non-collinear atoms in a fragment to construct the **fragment local coordinate frame** and we denote it as $\mathfrak{g}$. The fragment local frame is also equivariant to the rotation and translation of the input molecule. Similarly, we can obtain the rotation matrix $R_{\mathfrak{g} \to \mathfrak{w}}$ and translation vector $\boldsymbol{t}_{\mathfrak{g} \to \mathfrak{w}}$, which represent the orientation and displacement between the fragment local frame and the world frame, respectively.

Under the fragment local frame, we can obtain the local coordinates of atoms $V_{\mathcal{G}_i} \in \mathbb{R}^{q \times 3}$ in the $i$-th fragment, where $q$ denotes the number of atoms in a fragment. Then, we save each splited fragment from the training set into a dictionary with the key to be the canonical SMILES of each fragment and the value to be the atom types and atom local coordinates in each fragment under the associated fragment local frame.

With the transformation from the world frame to the molecule local frame and from the world frame to the fragment local frame, we can derive the transformation between the molecule local frame and the fragment local frame by using homogeneous transformation conversion. Specifically, we construct homogeneous transformation matrices from rotation matrices $R$ and translation vectors $\boldsymbol{t}$. Formally, we have

$$T_{\mathfrak{m} \to \mathfrak{w}} = \begin{bmatrix} R_{\mathfrak{m} \to \mathfrak{w}} & \boldsymbol{t}_{\mathfrak{m} \to \mathfrak{w}} \\ \boldsymbol{0} & 1 \end{bmatrix}, \quad T_{\mathfrak{g} \to \mathfrak{w}} = \begin{bmatrix} R_{\mathfrak{g} \to \mathfrak{w}} & \boldsymbol{t}_{\mathfrak{g} \to \mathfrak{w}} \\ \boldsymbol{0} & 1 \end{bmatrix}, \quad (2)$$

where $T \in \mathbb{R}^{4 \times 4}$ denotes the homogeneous transformation matrix and $\boldsymbol{0} \in \mathbb{R}^{3 \times 1}$ is a zero vector.

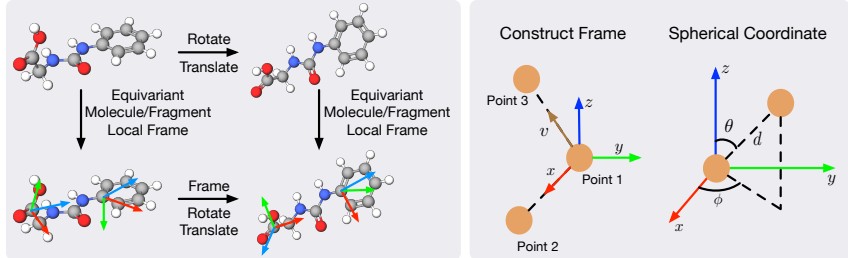

Figure 3: Illustraions of equivariant local frames. Left: If the molecule is translated and rotated, the molecule and fragment local frames are transformed accordingly. Right: Local frames are constructed using the first three non-collinear points, and the spherical coordinates are obtained under the local frame. More details are shown in Section 3.3.2 and Section 3.3.3.

Then, we can use the chain rule of coordinate frame transformations to obtain the homogeneous transformation between molecule and fragment local frames. Formally,

$$
\begin{aligned}
T_{\mathfrak{g}\to\mathfrak{m}} = T_{\mathfrak{m}\to\mathfrak{w}}^{-1} T_{\mathfrak{g}\to\mathfrak{w}} &= \begin{bmatrix} R_{\mathfrak{m}\to\mathfrak{w}}^T & -R_{\mathfrak{m}\to\mathfrak{w}}^T \boldsymbol{t}_{\mathfrak{m}\to\mathfrak{w}} \\ \mathbf{0} & 1 \end{bmatrix} \begin{bmatrix} R_{\mathfrak{g}\to\mathfrak{w}} & \boldsymbol{t}_{\mathfrak{g}\to\mathfrak{w}} \\ \mathbf{0} & 1 \end{bmatrix} \\
&= \begin{bmatrix} R_{\mathfrak{m}\to\mathfrak{w}}^T R_{\mathfrak{g}\to\mathfrak{w}} & R_{\mathfrak{m}\to\mathfrak{w}}^T (\boldsymbol{t}_{\mathfrak{g}\to\mathfrak{w}} - \boldsymbol{t}_{\mathfrak{m}\to\mathfrak{w}}) \\ \mathbf{0} & 1 \end{bmatrix}.
\end{aligned}
\tag{3}
$$

From the homogeneous transformation matrix $T_{\mathfrak{g}\to\mathfrak{m}}$, we can extract the rotation matrix $R_{\mathfrak{g}\to\mathfrak{m}}$ which will be used to obtain invariant representations as described in Section 3.3.3 and the translation vector $\boldsymbol{t}_{\mathfrak{g}\to\mathfrak{m}}$ such that

$$
R_{\mathfrak{g}\to\mathfrak{m}} = R_{\mathfrak{m}\to\mathfrak{w}}^T R_{\mathfrak{g}\to\mathfrak{w}}, \quad \boldsymbol{t}_{\mathfrak{g}\to\mathfrak{m}} = R_{\mathfrak{m}\to\mathfrak{w}}^T (\boldsymbol{t}_{\mathfrak{g}\to\mathfrak{w}} - \boldsymbol{t}_{\mathfrak{m}\to\mathfrak{w}}).
\tag{4}
$$

For each fragment in the fragment dictionary, apart from atom types and atom local coordinates, we also need to save the displacement $\boldsymbol{t}_{\mathfrak{g}\to c(\mathcal{G})}$ between the origin of the fragment local frame and the fragment center, which can be calculated via

$$
\boldsymbol{t}_{\mathfrak{g}\to c(\mathcal{G})} = V_{c(\mathcal{G})}^{\mathfrak{m}} - \boldsymbol{t}_{\mathfrak{g}\to\mathfrak{m}},
\tag{5}
$$

where $c(\mathcal{G})$ denotes the center of any fragment $\mathcal{G}$, and $V_{c(\mathcal{G})}^{\mathfrak{m}}$ represents the coordinates of the fragment center under the molecule local frame. $\boldsymbol{t}_{\mathfrak{g}\to c(\mathcal{G})}$ will be used when converting atom local coordinates from fragment local frame back to the world frame, which is described in Section 3.3.3.

### 3.3.3 $SE(3)$-INVARIANT FRAGMENT LOCAL REPRESENTATIONS

After establishing the frames, we use a function $f(\cdot)$ to convert the coordinates of each fragment center to **spherical coordinates** $d, \theta, \phi$ under the molecule frame $\mathfrak{m} = (\boldsymbol{x}, \boldsymbol{y}, \boldsymbol{z})$. Specifically, for each fragment center $\ell_i$ with coordinate $\boldsymbol{v}_{\ell_i}$, the corresponding spherical coordinate is

$$
\begin{aligned}
d_{\ell_i} &= \|\boldsymbol{v}_{\ell_i} - \boldsymbol{v}_{\ell_1}\|_2, \quad \theta_{\ell_i} = \arccos\left((\boldsymbol{v}_{\ell_i} - \boldsymbol{v}_{\ell_1}) \cdot \boldsymbol{z}/d_{\ell_i}\right), \\
\phi_{\ell_i} &= \operatorname{atan2}\left((\boldsymbol{v}_{\ell_i} - \boldsymbol{v}_{\ell_1}) \cdot \boldsymbol{y}, (\boldsymbol{v}_{\ell_i} - \boldsymbol{v}_{\ell_1}) \cdot \boldsymbol{x}\right).
\end{aligned}
\tag{6}
$$

The spherical coordinates show the relative position of each fragment under the molecule frame $\mathfrak{m}$. As shown in Figure 3, if the molecular coordinates are rotated by a matrix $R$ and translated by a vector $\boldsymbol{t}$, the transformed spherical coordinates remain the same, so the spherical coordinates are $SE(3)$-invariant. Compared to Cartesian coordinates, spherical coordinate values are bounded in a smaller region, namely, a range of $[0, \pi]/[0, 2\pi]$. Given the same numerical constraints, spherical coordinates require a smaller vocabulary size, and given the same vocabulary size, spherical coordinates present less information loss. This makes spherical coordinates advantageous in discretized representations and thus easier to be modeled by LMs. Experiments also show the superiority of invariant spherical coordinates over invariant Cartesian coordinates, as detailed in Appendix A.4

Apart from the spherical coordinates of the fragment center point under the molecule local frame, we also need the orientation of the fragment local frame with respect to the molecule local frame. One option is to directly use the rotation matrix $R_{\mathfrak{g}\to\mathfrak{m}}$ we derived above. However, the rotation

matrix representation has redundant information since 3D rotation in Euclidean space only has three degrees of freedom, and it can also unnecessarily increase the context length for LMs. To have a more compact rotation representation, we use a function $g(\cdot, \cdot)$ that takes in both the molecule and fragment local frames to obtain a rotation vector, which indicates the rotation axis and angle. Specifically, we first calculate the rotation matrix $R_{\mathfrak{g} \to \mathfrak{m}}$ in $g(\cdot, \cdot)$, as described in Section 3.3.2, then we can obtain the rotation angle $\psi$ and rotation axis $\boldsymbol{a} = (a_x, a_y, a_z)$ from the rotation matrix. Next, we can calculate **rotation vector** $\boldsymbol{m} = (m_x, m_y, m_z)$ via $\boldsymbol{m} = \psi \boldsymbol{a}$. Detailed derivation of $\psi$ and $\boldsymbol{a}$ can be found in Appendix B.

Conversely, given the spherical coordinates of the fragment centers under the molecule local frame and the rotation vector $\boldsymbol{m}$ between the fragment and molecule local frames, we can derive the atom coordinates in the world frame. Specifically, we first convert spherical coordinates to Cartesian coordinates $V_{c(\mathcal{G})}^{\mathfrak{m}}$ and transform the rotation vector back to the rotation matrix $R_{\mathfrak{g} \to \mathfrak{m}}$. Then, we can use coordinate transformation to convert atom coordinates from the fragment local frame to the world frame. Formally,

$$\boldsymbol{t}_{\mathfrak{g} \to \mathfrak{m}} = V_{c(\mathcal{G})}^{\mathfrak{m}} - \boldsymbol{t}_{\mathfrak{g} \to c(\mathcal{G})}, \quad V^{\mathfrak{m}} = V^{\mathfrak{g}} R_{\mathfrak{g} \to \mathfrak{m}}^T + \boldsymbol{t}_{\mathfrak{g} \to \mathfrak{m}}, \quad V^{\mathfrak{w}} = V^{\mathfrak{m}} R_{\mathfrak{m} \to \mathfrak{w}}^T + \boldsymbol{t}_{\mathfrak{m} \to \mathfrak{w}}. \tag{7}$$

where $V^{(\cdot)}$ denotes atom coordinates under a certain coordinate frame and $\boldsymbol{t}_{\mathfrak{g} \to c(\mathcal{G})}$ can be retrieved from the fragment dictionary built in Section 3.3.2. As for $R_{\mathfrak{m} \to \mathfrak{w}}$ and $\boldsymbol{t}_{\mathfrak{m} \to \mathfrak{w}}$, they are associated with the reference molecule for a given protein pocket when we construct the molecule local frame, and we can still use them when we generate new molecules for the same protein pocket.

Formally, with the $SE(3)$-equivariant local frames constructed in Section 3.3.2, we have the following properties of local representations built in this section.

**Lemma 3.3.** *Let $\mathcal{M} = (V, Z)$ be a 3D molecular graph with node type matrix $Z$ and node coordinate matrix $V$. Let $\mathfrak{m}$ be equivariant local frames of $\mathcal{M}$ built based on the first three non-collinear fragment centers in $\boldsymbol{L}(\mathcal{M})$ and $\mathfrak{g}$ be equivariant local frames of any fragment $\mathcal{G}_i$ built based on the first three non-collinear atoms in $\boldsymbol{L}(\mathcal{G}_i)$. $f(\cdot)$ is our function that maps 3D coordinate matrix $V$ of $\mathcal{M}$ to spherical representations $\boldsymbol{S}$ under the molecule local frame $\mathfrak{m}$. $g(\cdot, \cdot)$ is the function that maps the molecule local frame $\mathfrak{m}$ and fragment local frame $\mathfrak{g}$ to rotation vectors $\boldsymbol{m}$. Then for any 3D transformation $\tau \in SE(3)$, we have $f(V) = f(\tau(V))$ and $g(\mathfrak{m}, \mathfrak{g}) = g(\tau(\mathfrak{m}, \mathfrak{g}))$. Given spherical representations $\boldsymbol{S} = f(V)$ and rotation vectors $\boldsymbol{m} = g(\mathfrak{m}, \mathfrak{g})$, there exist a transformation $\tau \in SE(3)$, such that $f^{-1}(\boldsymbol{S}) = \tau(V)$ and $g^{-1}(\boldsymbol{m}) = \tau(\mathfrak{m}, \mathfrak{g})$.*

Lemma 3.3 shows that our spherical and rotation vector representations are $SE(3)$-invariant, and we can reconstruct original molecules from those representations, differing only by a $SE(3)$ transformation. Detailed proof is provided in Appendix C.2.

### 3.3.4 FRAG2SEQ: FRAGMENT AND GEOMETRY AWARE TOKENIZATION

With ordering that reduces molecule structures to 1D sequences and $SE(3)$-invariant spherical and rotation representations, both ensuring minimum 3D information loss, we develop a reversible transformation from 3D molecular fragments to 1D sequences. Figure 2 shows an overview of our method, which we term as Frag2Seq. Specifically, given a molecule $\mathcal{M}$ with $k$ fragments, we concatenate the fragment-position vector $[s_i, d_i, \theta_i, \phi_i, m_{xi}, m_{yi}, m_{zi}]$ of every fragment $\mathcal{G}_i$ in $\mathcal{M}$ into a 1D sequence by its canonical order, $\ell_1, \cdots, \ell_k$, where $s_i$ is the canonical SMILES string of $\mathcal{G}_i$. To define its properties, we formulate Frag2Seq and our major theoretical derivations below.

**Theorem 3.4.** *[Bijective Mapping] Following Def. 3.1, let $\mathcal{M}_1 = (V_1, Z_1)$ and $\mathcal{M}_2 = (V_2, Z_2)$ be two 3D molecular graphs. Let $\boldsymbol{L}(\mathcal{M})$ be the canonical order for $\mathcal{M}$ and $f(\cdot)$ and $g(\cdot, \cdot)$ be the functions following Lemma 3.3. Let the fragment-position vector for a molecule fragment $\mathcal{G}_i$ be $\boldsymbol{x}_i^* = [s_i, d_i, \theta_i, \phi_i, m_{xi}, m_{yi}, m_{zi}]$, where the vector elements are derived from $f(\cdot)$ and $g(\cdot, \cdot)$ as predefined. For the fragments $\{\mathcal{G}_i\}_{i=1}^k$ of $\mathcal{M}$, we construct canonical order $\boldsymbol{L}(\mathcal{M})$ for the $k$ fragments, $\ell_1, \cdots, \ell_k$, specifically by the order of a fragment's first-ranked atom. We define $Frag2Seq : \mathcal{M} \to \mathcal{U}$, which maps a molecule $\mathcal{M} \in \mathcal{M}$ to its sequence representation $U \in \mathcal{U}$, the set of all possible sequence representations, as*

$$Frag2Seq(\mathcal{M}) = concat(\boldsymbol{x}_{\ell_1}^*, \cdots, \boldsymbol{x}_{\ell_k}^*),$$

*where $concat(\cdot)$ concatenates elements as a sequence. Then $Frag2Seq(\cdot)$ is a surjective function, and the following equivalence holds:*

$$Frag2Seq(\mathcal{M}_1) = Frag2Seq(\mathcal{M}_2) \Leftrightarrow \mathcal{M}_1 \cong_{3D} \mathcal{M}_2,$$

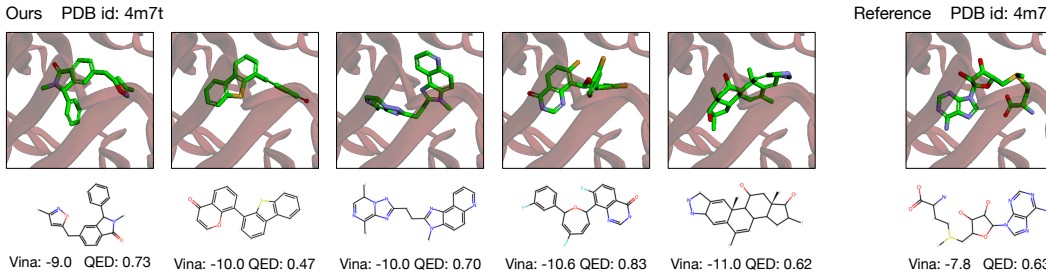

Figure 4: Visualization of our generated 3D molecules that have higher binding affinity than the reference molecule in the test set. A higher QED indicates more drug-likeness, and a lower Vina score indicates higher binding affinity.

*where $\mathcal{M}_1 \cong_{3D} \mathcal{M}_2$ denotes $\mathcal{M}_1$ and $\mathcal{M}_2$ are 3D isomorphic. If we allow rounding up spherical coordinate and rotation vector values to $\geq b$ decimal places, then the surjectivity and equivalence still hold, only molecules $\mathcal{M}_1$ and $\mathcal{M}_2$ are $(|10^{-b}|/2)$-constrained 3D isomorphic.*

Theorem 3.4 establishes guarantees that we can uniquely construct a 1D sequence given a 3D molecule using Frag2Seq, and uniquely reconstruct a 3D molecule given a sequence output of Frag2Seq. Furthermore, two sequence outputs from Frag2Seq are identical if and only if the two corresponding molecules are 3D isomorphic. This enables sequential tokenization of 3D molecules while preserving structural completeness and geometric invariance. Due to the necessity of discreteness in tokenization for LMs, in reality, numerical values need to be discretized before concatenation. In practice, we round up numerical values to certain decimal places. In this case, Theorem 3.4 also extends guarantees for the practical use of Frag2Seq. If we allow a round-up error below $|10^{-b}|/2$ for coordinates when distinguishing 3D isomorphism, all properties still hold. This implies that the practical Frag2Seq implementation retains near-complete geometric information and invariance, with numerical precision of $\epsilon \leq |10^{-b}|/2$.

With discreteness incorporated, we can collect a finite vocabulary covering all accessible fragment samples to enable tokenization for LMs. Specifically, we use vocabularies consisting of fragment type tokens such as *O, O=C, c1ccccc1,* $\cdots$, and numerical tokens such as $-8.126, 0.05°$ or $-1.702^*$. The numerical tokens range from the smallest to the largest distance, angle, and rotation values with restricted precision of 2 or 3 decimal places.

## 3.4 CONDITIONAL TRAINING AND GENERATION

After we obtain the sequence representation of 3D molecules $U = \{u_1, \cdots, u_n\}$, we need to model the distribution of such 3D geometry-aware sequences. We adopt GPT (Radford et al., 2018) as the base model to learn the distribution. To incorporate the protein pocket condition, we use pre-trained inverse folding model ESM-IF1 (Hsu et al., 2022) to obtain node embeddings of the protein backbone. Then, we add cross-attention between protein node embeddings and ligand token embeddings after the multi-head self-attention in each attention block, in which queries are from ligand token embedding and keys and values are from protein node embedding. For training, we use the standard next-token prediction cross-entropy loss to maximize the following likelihood:

$$\mathcal{L}(U; \mathcal{P}) = \sum_i \log p_\theta(u_i | u_{i-1}, \cdots, u_1; \mathcal{P}). \tag{8}$$

To generate molecules conditioned on a protein pocket, we first need to sample initial fragment tokens from the first-token distribution in the training set. Then, we can generate the following tokens autoregressively by sampling from the conditional distribution $p_\theta(u_i | u_{i-1}, \cdots, u_1; \mathcal{P})$ and we stop the generation when the maximum length is reached or the ending token is sampled.

## 4 EXPERIMENTS

We empirically show the effectiveness of our method in generating ligands for protein pockets. In Section 4.1, we describe the experimental setup, including dataset, baselines, and evaluation metrics. Then, in Section 4.2, we present our results about binding affinity, chemical properties, and efficiency. In Section 4.3, we show more results about structural analysis. Due to the limited space, we conduct ablation studies on several design choices in Appendix A.4, and more results about efficiency in Appendix A.5.

Table 1: Summary of molecular properties for generated molecules by our method and other baselines. * denotes the property results taken from the DiffSBDD paper. † denotes that we re-evaluate the properties of generated molecules of those baselines using the DiffSBDD evaluation code for a fair comparison. The best results are annotated in bold.

| Methods | Vina Score (↓) | High Affinity (↑) | QED (↑) | SA (↑) | Lipinski (↑) | Diversity (↑) | Time (s, ↓) |
|---|---|---|---|---|---|---|---|
| Test set | $-6.871 \pm 2.32$ | $-$ | $0.476 \pm 0.20$ | $0.728 \pm 0.14$ | $4.340 \pm 1.14$ | $-$ | $-$ |
| 3D-SBDD* | $-5.888 \pm 1.91$ | $0.364 \pm 0.31$ | $0.502 \pm 0.17$ | $0.675 \pm 0.14$ | $4.787 \pm 0.51$ | $0.742 \pm 0.09$ | $15986.4 \pm 9851.0$ |
| Pocket2Mol* | $-7.058 \pm 2.80$ | $0.515 \pm 0.31$ | $0.572 \pm 0.16$ | $\mathbf{0.752 \pm 0.12}$ | $4.936 \pm 0.27$ | $0.735 \pm 0.15$ | $2827.3 \pm 1456.8$ |
| GraphBP* | $-4.719 \pm 4.03$ | $0.183 \pm 0.21$ | $0.502 \pm 0.12$ | $0.307 \pm 0.09$ | $4.883 \pm 0.37$ | $\mathbf{0.844 \pm 0.01}$ | $1162.8 \pm 438.5$ |
| TargetDiff* | $-7.318 \pm 2.47$ | $0.581 \pm 0.31$ | $0.483 \pm 0.20$ | $0.584 \pm 0.13$ | $4.594 \pm 0.83$ | $0.718 \pm 0.09$ | $\sim 3428$ |
| DecompDiff† | $-6.607 \pm 2.11$ | $0.423 \pm 0.25$ | $0.496 \pm 0.21$ | $0.659 \pm 0.14$ | $4.493 \pm 1.02$ | $0.722 \pm 0.10$ | $\sim 6189$ |
| DiffSBDD* | $-7.177 \pm 3.28$ | $0.499 \pm 0.30$ | $0.556 \pm 0.20$ | $0.729 \pm 0.12$ | $4.742 \pm 0.59$ | $0.718 \pm 0.07$ | $629.9 \pm 277.2$ |
| FLAG† | $-6.389 \pm 3.24$ | $0.478 \pm 0.34$ | $0.487 \pm 0.19$ | $0.702 \pm 0.15$ | $4.656 \pm 0.74$ | $0.701 \pm 0.14$ | $1289.1 \pm 378.0$ |
| DrugGPS† | $-6.608 \pm 2.38$ | $0.421 \pm 0.24$ | $0.467 \pm 0.21$ | $0.628 \pm 0.15$ | $4.495 \pm 0.99$ | $0.738 \pm 0.10$ | $1007.8 \pm 554.1$ |
| Lingo3DMol† | $-7.257 \pm 1.69$ | $0.625 \pm 0.36$ | $0.269 \pm 0.15$ | $0.656 \pm 0.08$ | $3.121 \pm 1.25$ | $0.480 \pm 0.12$ | $1481.9 \pm 1512.8$ |
| Frag2Seq | $\mathbf{-7.366 \pm 1.96}$ | $\mathbf{0.653 \pm 0.33}$ | $\mathbf{0.645 \pm 0.15}$ | $0.642 \pm 0.11$ | $\mathbf{4.989 \pm 0.11}$ | $0.711 \pm 0.07$ | $\mathbf{48.8 \pm 14.6}$ |

## 4.1 SETUP

**Dataset.** Following Schneuing et al. (2022), we use the CrossDocked dataset (Francoeur et al., 2020) that is further curated by previous work (Luo et al., 2021). More details about the dataset is provided in Appendix A.1.

**Baselines.** We compare with recent generation methods in structure-based drug design. 3D-SBDD (Luo et al., 2021), Pocket2Mol (Peng et al., 2022), and GraphBP (Liu et al., 2022a) generate atoms in an autoregressive scheme. TargetDiff (Guan et al., 2022), DecompDiff (Guan et al., 2023), and DiffSBDD (Schneuing et al., 2022) are diffusion-based methods that generate all atoms in one shot. FLAG (Zhang et al., 2022) and DrugGPS (Zhang & Liu, 2023) generate molecules motif-by-motif. Lingo3DMol (Feng et al., 2024) uses a language model to generate fragment-based SMILES and then predict coordinates. The language model is pre-trained on 12 million 3D molecules using a denoising strategy.

**Evaluation Metrics.** We adopt commonly-used metrics used in previous work (Schneuing et al., 2022; Peng et al., 2022) to evaluate the quality of our generated molecules: (1) **Vina Score** estimates the binding affinity between generated molecules and given protein pockets; (2) **High Affinity** measures the percentage of generated molecules that have higher binding affinity than the reference molecule for a certain protein; (3) **QED** is a measure used to assess the drug-likeness of a molecule based on its molecular properties; (4) **SA** estimates how easy it would be to synthesize a given chemical compound; (5) **Lipinski** measures how well a molecule satisfies the Lipinski's rule of five (Lipinski et al., 2012), which evaluates the drug-likeness of a molecule; (6) **Diversity** measures the average pairwise diversity (calculated as $1 - \text{Tanimoto similarity}$) of generated molecules for a binding pocket; (7) **Time** is the average time cost to generate 100 molecules for a protein pocket in the test set. All the Vina scores are calculated by QVina (Alhossary et al., 2015), and the chemical properties are calculated by RDKit (Bento et al., 2020).

## 4.2 RESULTS

We show the results of the above metrics for our method and baselines in Table 1. The results show that our method can achieve better or competitive performance with other baseline methods. Specifically, our method achieves the best result on QED and Lipinski, which indicates that our generated molecules possess better drug characteristics. Additionally, our method obtains the best Vina score and high affinity metric, which shows that our model captures more accurate interactions between ligands and protein pockets to improve the binding affinity. Moreover, our method has much better sampling efficiency than the baselines due to our simple generation pipeline and fragment-based generation strategy.

In Figure 4, we provide several examples of our generated 3D molecules for a certain protein pocket (PDB id 4m7t). The reference molecule is provided in the test set, and our generated molecules exhibit higher binding affinity than the reference molecule. These examples can validate that our method has the capability to capture the complex interaction between proteins and ligands in order

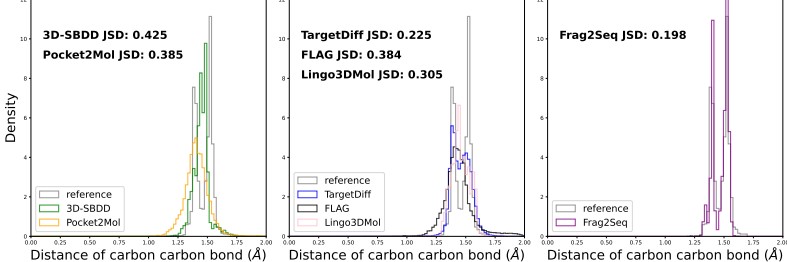

Figure 5: Comparison of the distribution of carbon-carbon bond distance. Jensen-Shannon divergence (JSD) between reference molecules and generated molecules is reported.

to generate novel molecules that have higher binding affinity while retaining similar or even better drug-likeness properties than the reference molecule.

### 4.3 STRUCTURAL ANALYSIS

In Figure 5, we plot the empirical distribution of carbon-carbon bond distances of generated molecules and reference molecules. The reference distribution (gray) has two modes, but most methods can only capture one mode due to mode collapse. Even though TargetDiff exhibits two modes in the distribution, it still suffers from the over-smoothness issue. Comparatively, our method can better capture those two modes in the reference distribution. Furthermore, we calculate Jensen-Shannon divergence (JSD) (Lin, 1991) between the bond distance distribution of reference and generated molecules for each method to quantitatively evaluate the distribution fitting capability. We can observe that our method outperforms other methods by a clear margin.

To further evaluate the structures of generated molecules, we calculate the Kullback-Leibler (KL) divergence of bond angles and dihedral angles between generated molecules and reference molecules in the test set. The results in Table 2 show that our method achieves better KL divergence than other representative baselines in most cases, which demonstrates that Frag2Seq can well capture the structural distribution of the data and generate more valid substructures.

Table 2: The KL divergence of the distribution of the bond and dihedral angles between generated molecules and reference molecules. The lower letters indicate the atoms in the aromatic rings. * denotes numbers taken from the original paper due to the absence of those dihedral angles in regenerated structures.

| Angles | liGAN | 3D-SBDD | Pocket2Mol | TargetDiff | FLAG | Frag2Seq |
|--------|-------|---------|------------|------------|------|----------|
| CCC | 1.266 | 0.465 | 0.770 | 0.507 | 0.497 | **0.263** |
| CCO | 1.475 | 0.618 | 1.104 | 0.713 | 0.768 | **0.306** |
| NCC | 1.312 | 0.619 | 0.754 | 0.585 | **0.427** | 0.578 |
| CCCC | 0.197 | 0.250 | 0.215 | 0.170 | 0.164 | **0.113** |
| cccc | 0.786 | 1.044 | 4.49* | 0.229 | 0.516 | **0.041** |
| CCCO | 0.206 | 0.263 | 0.245 | 0.191 | **0.180** | 0.198 |
| Cccc | 0.725 | 0.939 | 2.85* | 0.397 | 0.311 | **0.109** |
| CC=CC | 0.526 | 0.655 | 0.296 | 0.301 | 0.345 | **0.133** |

## 5 CONCLUSIONS, BROADER IMPACTS, AND LIMITATIONS

In this work, we develop a method to convert molecules into fragment-based $SE(3)$-invariant sequences which can be naturally processed by language models. We use a protein inverse folding model to obtain protein embeddings and integrate them into the language model via cross-attention. Our experiments show the great potential of using language models in structure-based drug design, which can potentially accelerate drug discovery by enhancing the efficiency and accuracy of identifying viable drug candidates. However, our method also has limitations. The chemical bonds are not directly generated by the language model and need to be inferred by OpenBabel (O'Boyle et al., 2011), and real number tokenization requires quantization, leading to some accuracy loss.

## ACKNOWLEDGMENTS

This work was supported partially by National Science Foundation grant IIS-2243850 and National Institutes of Health grant U01AG070112 (to S.J.), and by the Molecule Maker Lab Institute (to H.J.): an AI research institute program supported by NSF under award No. 2019897. The views and conclusions contained herein are those of the authors and should not be interpreted as necessarily representing the official policies, either expressed or implied, of the U.S. Government. The U.S. Government is authorized to reproduce and distribute reprints for governmental purposes notwithstanding any copyright annotation therein.

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

# A   EXPERIMENTAL DETAILS AND ADDITIONAL RESULTS

## A.1   DATASET DETAILS

Following previous work in SBDD (Schneuing et al., 2022; Luo et al., 2021; Peng et al., 2022; Guan et al., 2022; 2023), we use the CrossDocked2020 dataset to train and test our model. The dataset was originally created by Francoeur et al. (2020). We adopt the version that is further filtered in the previous work (Luo et al., 2021). To increase the data quality, they filter out data points that have binding pose RMSD greater than 1Å. The data split is performed according to 30% sequence identity using MMseqs2 (Steinegger & Söding, 2017). Finally, there are 100000 protein-ligand training pairs and 100 proteins in the test set. The dataset curated by Luo et al. (2021) is publicly available at `https://github.com/luost26/3D-Generative-SBDD/blob/main/data/README.md` under the MIT license.

## A.2   PROTEIN ENCODER DETAILS

We use the pre-trained ESM-IF1 model to obtain protein node embeddings. ESM-IF1 model is also known as GVP-Transformer that is developed by Hsu et al. (2022). It is an inverse folding model that is used to design protein amino acid sequences given 3D backbone structures. The ESM-IF1 model uses a modified Geometric Vector Perceptron (GVP) layers (Jing et al., 2020) to extract rotational and translational invariant features, followed by an autoregressive encoder-decoder Transformer (Vaswani et al., 2017). The model weights of ESM-IF1 are publicly available at `https://github.com/facebookresearch/esm` under the MIT license.

## A.3   NETWORK AND TRAINING DETAILS

For the language model, we use the GPT-1 architecture (Radford et al., 2018) with 12 layers, 12 attention heads, and a hidden embedding size of 768. Cross-attention is added after self-attention in each layer. The protein embedding size generated by the ESM-IF1 model is $512$ and is increased to 768 by an MLP layer before the protein embeddings are fed into the cross-attention. The architecture of the cross-attentions is the same as the self-attention, except that the key and value are from the protein embeddings. The initial learning rate is set as $4e^{-4}$, and the batch size is 64. We implement our methods using PyTorch (Paszke et al., 2019) and we adopt the AdamW optimizer (Loshchilov & Hutter, 2017) with $\beta_1 = 0.9$, $\beta_2 = 0.95$. We adopt linear warm up and cosine scheduler to adaptively adjust the learning rate. Specifically, during the first $10\%$ of total training tokens, we increase the learning rate from 0 to the initial learning rate. Then, the learning rate decreases to $4e^{-5}$ by using a cosine decay scheduler. We use a single NVIDIA A100 GPU to train our model. We set the maximum context length as $512$ for all the experiments except for the ablation study on converting proteins into sequences. In that case, the maximum context length is increased to $850$ to accommodate the preappended protein sequences.

## A.4   ABLATION STUDY

In this section, we conduct ablations studies on the 3D representation, fragmentation, and protein embedding in order to show how these factors affect the performance of the language model on the SBDD task.

First, we compare the effect of different 3D representations of fragment center under $SE(3)$-equivariant molecule local frame proposed in Section 3.3.2. We explore the use of $SE(3)$-invariant Cartesian coordinates instead of $SE(3)$-invariant spherical coordinates used in Section 3.3.3. From the result in Table 3, we can see that using spherical coordinates can achieve better binding affinity and chemical properties. By using spherical coordinates, the distances and angles are constrained to a smaller region, which makes the language model easier to learn structure correlations.

Then, we show that fragment-based tokenization is crucial to achieving better generation results. Specifically, we try to generate molecules atom-by-atom instead of using fragments. In this setting, we do not need to build fragment local frames, and rotation vectors do not exist in converted sequences. Results in Table 4 show that fragment-based generation can achieve much better performance on binding affinity and most drug-likeness properties, which validates the significance of fragmentation.

Finally, we compare different ways to incorporate protein pocket information into the model. Specifically, we convert protein structures to sequences in the same way as atom-based tokenization of small molecules. Then, we preappend the tokenized protein sequence to the corresponding ligand sequence and feed it into the language model. Results in Table 5 show that extracting protein node embeddings and integrating them into the language model via cross-attention can more effectively capture the interaction between proteins and ligands.

Table 3: Generation performance between using invariant Cartesian coordinates and spherical coordinates representations of fragment centers.

| Methods | Vina Score (↓) | QED (↑) | SA (↑) | Lipinski (↑) | Diversity (↑) |
|---|---|---|---|---|---|
| Frag2Seq-Invariant Coordinates | $-7.242 \pm 2.11$ | $0.636 \pm 0.16$ | $0.637 \pm 0.10$ | $4.974 \pm 0.17$ | $\mathbf{0.714 \pm 0.09}$ |
| Frag2Seq-Spherical Coordinates | $\mathbf{-7.366 \pm 1.96}$ | $\mathbf{0.645 \pm 0.15}$ | $\mathbf{0.642 \pm 0.11}$ | $\mathbf{4.989 \pm 0.11}$ | $0.711 \pm 0.07$ |

Table 4: Generation performance between using atom-by-atom and fragment-by-fragment manners.

| Methods | Vina Score (↓) | QED (↑) | SA (↑) | Lipinski (↑) | Diversity (↑) |
|---|---|---|---|---|---|
| Frag2Seq-Atom | $-6.895 \pm 1.42$ | $0.484 \pm 0.22$ | $\mathbf{0.683 \pm 0.11}$ | $4.677 \pm 0.50$ | $\mathbf{0.737 \pm 0.19}$ |
| Frag2Seq-Fragment | $\mathbf{-7.366 \pm 1.96}$ | $\mathbf{0.645 \pm 0.15}$ | $0.642 \pm 0.11$ | $\mathbf{4.989 \pm 0.11}$ | $0.711 \pm 0.07$ |

Table 5: Generation performance between different ways to incorporate protein information.

| Methods | Vina Score (↓) | QED (↑) | SA (↑) | Lipinski (↑) | Diversity (↑) |
|---|---|---|---|---|---|
| Frag2Seq-Protein Seq | $-6.646 \pm 1.64$ | $0.605 \pm 0.17$ | $\mathbf{0.691 \pm 0.13}$ | $4.968 \pm 0.18$ | $0.493 \pm 0.39$ |
| Frag2Seq-ESM-IF1 | $\mathbf{-7.366 \pm 1.96}$ | $\mathbf{0.645 \pm 0.15}$ | $0.642 \pm 0.11$ | $\mathbf{4.989 \pm 0.11}$ | $\mathbf{0.711 \pm 0.07}$ |

## A.5 GENERATION EFFICIENCY ANALYSIS

In this section, we compare the generation efficiency of our method and other representative baseline methods. We use a single NVIDIA 2080Ti GPU and a batch size of 10. The results in Table 6 show that our method exhibits better sampling efficiency than other autoregressive and diffusion-based models.

Table 6: Generation efficiency comparison among autoregressive methods, diffusion methods, and our fragment-based LM method.

| Methods | Parameters | Memory | Sample/second |
|---|---|---|---|
| 3D-SBDD | 1.2M | 3.4GB | 0.005 |
| Pocket2Mol | 3.7M | 1.2GB | 0.008 |
| DrugGPS | 5.1M | 2.5GB | 0.73 |
| TargetDiff | 2.8M | 1.8GB | 0.01 |
| Frag2Seq | 134.3M | 2.2GB | 2.0 |

## A.6 PARAMETER SIZES AND PERFORMANCE COMPARISON

In this section, we compare different parameter sizes of the language models (130M, 70M, 40M), as shown in Table 7. Drug-likeness properties exhibit similar performance across the three models, but there is a noticeable degradation in binding energy as the model size decreases. We hypothesize this is due to the inherent nature of language models, which require a large number of parameters to learn intricate interaction patterns, which is required in learning interactions between proteins and molecules. However, even with large parameter sizes, the language model still offers superior sampling efficiency compared to previous methods, as compared in Table 6.

Table 7: Generation performance between different sizes of language models.

| Methods | Vina Score ($\downarrow$) | QED ($\uparrow$) | SA ($\uparrow$) | Lipinski ($\uparrow$) | Diversity ($\uparrow$) |
|---|---|---|---|---|---|
| Frag2Seq-130M | $-\mathbf{7.366} \pm \mathbf{1.96}$ | $\mathbf{0.645} \pm \mathbf{0.15}$ | $0.642 \pm 0.11$ | $\mathbf{4.989} \pm \mathbf{0.11}$ | $0.711 \pm 0.07$ |
| Frag2Seq-70M | $-7.255 \pm 1.83$ | $0.630 \pm 0.16$ | $0.640 \pm 0.11$ | $4.977 \pm 0.16$ | $\mathbf{0.718} \pm \mathbf{0.08}$ |
| Frag2Seq-40M | $-7.228 \pm 2.09$ | $0.643 \pm 0.16$ | $\mathbf{0.645} \pm \mathbf{0.11}$ | $4.982 \pm 0.14$ | $0.708 \pm 0.07$ |

### A.7 COMPARISON WITH IPDIFF

In this section, we compare our method with IPDiff, another diffusion model for structure-based drug design. We do not include this comparison in the main text because the training data resources are not exactly the same. Specifically, IPDiff leverages additional binding affinity data from PDBBind to pretrain a binding affinity prediction model, IPNet. This allows IPDiff to utilize protein-ligand interaction priors learned from IPNet, enabling the generation of molecules with improved target binding affinity and other properties.

We compare IPDiff with and without binding affinity pretraining, as shown in Table 8. To fairly compare with the non-pretrained version of IPDiff, we use the same energy evaluation code as IPDiff to assess our method. This yields slightly better results than those obtained using the DiffSBDD evaluation we initially adopted. Our results show that Frag2Seq outperforms IPDiff when IPDiff is not pretrained on binding affinity data and achieves comparable Vina scores to the pretrained version of IPDiff, despite Frag2Seq not utilizing any additional binding affinity data. This further validates the effectiveness of our approach. Moreover, Frag2Seq surpasses IPDiff across all drug-likeness metrics and exhibits a generation speed approximately 64 times faster than IPDiff.

However, IPDiff still demonstrates that incorporating a binding affinity prior can successfully enhance binding energy during generation. Thus, integrating a binding affinity prior into the language model is a promising direction for further improving Frag2Seq.

Table 8: Generation performance comparison with IPDiff. IPDiff results are taken from its paper. *We use the same energy evaluation code as IPDiff to evaluate Frag2Seq for fair comparison.

| Methods | Vina Score* ($\downarrow$) | QED ($\uparrow$) | SA ($\uparrow$) | Time (s, $\downarrow$) |
|---|---|---|---|---|
| IPDiff w/ pretrain on binding affinity data | $-\mathbf{8.57}$ | $0.46$ | $0.57$ | $3063$ |
| IPDiff w/o pretrain on binding affinity data | $-7.55$ | $0.52$ | $0.61$ | - |
| Frag2Seq | $-8.49*$ | $\mathbf{0.645} \pm \mathbf{0.15}$ | $\mathbf{0.642} \pm \mathbf{0.11}$ | $\mathbf{48.8} \pm \mathbf{14.6}$ |

## B DERIVATION OF ROTATION VECTOR BETWEEN FRAGMENT AND MOLECULE FRAMES

In Section 3.3.2, we obtain the rotation matrix $R_{\mathfrak{g}\to\mathfrak{m}}$ between fragment and molecule local frames. Then, we can derive the more compact rotation vector representation. Specifically, we can obtain the rotation angle $\psi$ by

$$\psi = cos^{-1}\left(\frac{tr(R_{\mathfrak{g}\to\mathfrak{m}}) - 1}{2}\right), \tag{9}$$

where $tr(\cdot)$ denote the trace of a matrix. Suppose $R_{\mathfrak{g}\to\mathfrak{m}}$ has the following form

$$R_{\mathfrak{g}\to\mathfrak{m}} = \begin{bmatrix} r_{11} & r_{12} & r_{13} \\ r_{21} & r_{22} & r_{23} \\ r_{31} & r_{32} & r_{33} \end{bmatrix}, \tag{10}$$

the rotation axis $\boldsymbol{a} = (a_x, a_y, a_z)$ can be obtained by

$$a_x = (r_{32} - r_{23})/2\sin\psi, \tag{11}$$
$$a_y = (r_{13} - r_{31})/2\sin\psi, \tag{12}$$
$$a_z = (r_{21} - r_{12})/2\sin\psi. \tag{13}$$

Then, we can calculate rotation vector $\boldsymbol{m} = (m_x, m_y, m_z)$ via $\boldsymbol{m} = \psi\boldsymbol{a}$.

# C  PROOFS

## C.1  PROOF OF LEMMA 3.2

**Lemma** (Canonical Ordering for 3D Molecular Graph Isomorphism). *Let $\mathcal{M}_1 = (V_1, Z_1)$ and $\mathcal{M}_2 = (V_2, Z_2)$ be two 3D molecular graphs following Def. 3.1. Let $\boldsymbol{L} : \mathcal{M} \to \mathcal{L}$ be a function that maps a molecule $\mathcal{M} \in \mathcal{M}$, the set of all finite 3D molecular graphs, to its canonical order $\boldsymbol{L}(\mathcal{M}) \in \mathcal{L}$, the set of all possible canonical orders, as produced by the canonical SMILES. Then the following equivalence holds:*

$$\boldsymbol{L}(\mathcal{M}_1) = \boldsymbol{L}(\mathcal{M}_2) \Leftrightarrow \mathcal{M}_1 \cong_{3D} \mathcal{M}_2$$

*where $\mathcal{M}_1 \cong_{3D} \mathcal{M}_2$ denotes that $\mathcal{M}_1$ and $\mathcal{M}_2$ are 3D isomorphic.*

*Proof.* We first prove the equivalence from right to left.

Given that $\mathcal{M}_1 \cong_{3D} \mathcal{M}_2$, from Def. 3.1, there exists a bijection $b : ver(\mathcal{M}_1) \to ver(\mathcal{M}_2)$ such that for every atom in $\mathcal{M}_1$ indexed $i$, $\boldsymbol{z}_i^{\mathcal{M}_1} = \boldsymbol{z}_{b(i)}^{\mathcal{M}_2}$, and there exists a 3D transformation $\tau \in SE(3)$ such that $\boldsymbol{v}_i^{\mathcal{M}_1} = \tau(\boldsymbol{v}_{b(i)}^{\mathcal{M}_2})$. Therefore, $\mathcal{M}_2$ is a rotated and translated instance of $\mathcal{M}_1$. In this case, $\mathcal{M}_1$ and $\mathcal{M}_2$ has identical molecular structure. Canonical SMILES (O'Boyle, 2012; Weininger et al., 1989) provides a unique string representation for a given molecular structure. We leave out the proof for the rigor of the canonical SMILES algorithm, which has been provided by O'Boyle (2012). Thus for the canonical SMILES string of $\mathcal{M}_1$ and $\mathcal{M}_2$, $s_1$ and $s_2$, we have $s_1 = s_2$. Since $\boldsymbol{L}(\mathcal{M}_1)$ and $\boldsymbol{L}(\mathcal{M}_2)$ are the ordering in $s_1$ and $s_2$, respectively, we have $\boldsymbol{L}(\mathcal{M}_1) = \boldsymbol{L}(\mathcal{M}_2)$.

Next we prove the equivalence from left to right. For two 3D molecular graphs $\boldsymbol{L}(\mathcal{M}_1)$ and $\boldsymbol{L}(\mathcal{M}_2)$, we know $\boldsymbol{L}(\mathcal{M}_1) = \boldsymbol{L}(\mathcal{M}_2)$. According to the canonical SMILES properties, if two molecules are not identical in structure, they are guaranteed to receive different canonical SMILES strings. Therefore, since $\boldsymbol{L}(\mathcal{M}_1) = \boldsymbol{L}(\mathcal{M}_2)$, $\mathcal{M}_1$ and $\mathcal{M}_2$ must be identical in structure. Then there exists bijection $b : ver(\mathcal{M}_1) \to ver(\mathcal{M}_2)$ such that $\boldsymbol{z}_i^{\mathcal{M}_1} = \boldsymbol{z}_{b(i)}^{\mathcal{M}_2}$ for every atom in $\mathcal{M}_1$ indexed $i$. Meanwhile, in this case, $\mathcal{M}_2$ can be derived from $\mathcal{M}_1$ after certain homogeneous 3D transformation under the right-handed Cartesian coordinate system, *i.e.*, $SE(3)$ transformation. An arbitrary rigid transformation in $SE(3)$ can be separated into two parts, a translation and a rigid rotation. Let the translation and rotation needed be represented as $\tau_1, \tau_2$. $SE(3)$ is a Lie group and satisfies the axioms that the set is closed under the binary operation. $\tau_1, \tau_2 \in SE(3)$, thus $\tau_1 \tau_2 \in SE(3)$. Let $\tau' = \tau_1 \tau_2 \in SE(3)$. Therefore, there exist a 3D transformation $\tau' \in SE(3)$ such that $\boldsymbol{v}_i^{\mathcal{M}_1} = \tau'(\boldsymbol{v}_{b(i)}^{\mathcal{M}_2})$. Consequently, $\mathcal{M}_1 \cong_{3D} \mathcal{M}_2$.

This completes the proof.  □

## C.2  PROOF OF LEMMA 3.3

**Lemma.** *Let $\mathcal{M} = (V, Z)$ be a 3D molecular graph with node type vector $\boldsymbol{z}$ and node coordinate matrix $V$. Let $\mathfrak{m}$ be equivariant local frames of $\mathcal{M}$ built based on the first three non-collinear fragment centers in $\boldsymbol{L}(\mathcal{M})$ and $\mathfrak{g}$ be equivariant local frames of any fragment $\mathcal{G}_i$ built based on the first three non-collinear atoms in $\boldsymbol{L}(\mathcal{G}_i)$. $f(\cdot)$ is our function that maps 3D coordinate matrix $V$ of $\mathcal{M}$ to spherical representations $\boldsymbol{S}$ under the molecule local frame $\mathfrak{m}$. $g(\cdot, \cdot)$ is the function that maps the molecule local frame $\mathfrak{m}$ and fragment local frame $\mathfrak{g}$ to rotation vectors $\boldsymbol{m}$. Then for any 3D transformation $\tau \in SE(3)$, we have $f(V) = f(\tau(V))$ and $g(\mathfrak{m}, \mathfrak{g}) = g(\tau(\mathfrak{m}, \mathfrak{g}))$. Given spherical representations $\boldsymbol{S} = f(V)$ and rotation vectors $\boldsymbol{m} = g(\mathfrak{m}, \mathfrak{g})$, there exist a transformation $\tau \in SE(3)$, such that $f^{-1}(\boldsymbol{S}) = \tau(V)$ and $g^{-1}(\boldsymbol{m}) = \tau(\mathfrak{m}, \mathfrak{g})$.*

*Proof.* We first prove the $SE(3)$-invariance of the spherical representations of fragment center under molecule local frame, then prove the rotation representations between the fragment and molecule local frames are also $SE(3)$-invariant.

To simplify the proof, we assume that the molecule is centered first by moving the first atom to the origin. Let $\ell_1, \ell_2$, and $\ell_m$ be the indices of the first three non-collinear fragment centers in $\mathcal{M}$, and

the molecule local frame $\mathbb{m} = (\boldsymbol{x}, \boldsymbol{y}, \boldsymbol{z})$ is calculated as

$$
\begin{aligned}
\boldsymbol{x} &= \text{normalize}(\boldsymbol{v}_{\ell_2} - \boldsymbol{v}_{\ell_1}), \\
\boldsymbol{y} &= \text{normalize}\left((\boldsymbol{v}_{\ell_m} - \boldsymbol{v}_{\ell_1}) \times \boldsymbol{x}\right), \\
\boldsymbol{z} &= \boldsymbol{x} \times \boldsymbol{y},
\end{aligned}
\tag{14}
$$

where normalize$(\cdot)$ denotes normalizing input vectors into unit vectors. Given any $SE(3)$ transformation $\tau$ containing rotation matrix $R$ and translation vector $\boldsymbol{t}$, we have transformed coordinate $\boldsymbol{v}_i' = \tau(\boldsymbol{v}_i) = R\boldsymbol{v}_i + \boldsymbol{t}$. Then, we can apply this transformation to the vectors in Equation (14) to obtain the transformed basis vectors for the molecule local frame. Specifically,

$$
\begin{aligned}
\boldsymbol{x}' &= \text{normalize}(R\boldsymbol{v}_{\ell_2} + \boldsymbol{t} - (R\boldsymbol{v}_{\ell_1} + \boldsymbol{t})) = R(\text{normalize}(\boldsymbol{v}_{\ell_2} - \boldsymbol{v}_{\ell_1})) = R\boldsymbol{x}, \\
\boldsymbol{y}' &= \text{normalize}\left((R\boldsymbol{v}_{\ell_m} + \boldsymbol{t} - (R\boldsymbol{v}_{\ell_1} + \boldsymbol{t})) \times \boldsymbol{x}'\right), \\
&= \text{normalize}\left(R\left((\boldsymbol{v}_{\ell_m} - \boldsymbol{v}_{\ell_1}) \times \boldsymbol{x}\right)\right), \\
&= R(\text{normalize}\left((\boldsymbol{v}_{\ell_m} - \boldsymbol{v}_{\ell_1}) \times \boldsymbol{x}\right)), \\
&= R\boldsymbol{y}, \\
\boldsymbol{z}' &= \boldsymbol{x}' \times \boldsymbol{y}' = R\boldsymbol{x} \times R\boldsymbol{y} = R(\boldsymbol{x} \times \boldsymbol{y}) = R\boldsymbol{z}.
\end{aligned}
\tag{15}
$$

So, the new molecule local frame is subject to the same transformation. Then, for spherical coordinates of any fragment center $f(V)_{\ell_i}$ that is represented as

$$
\begin{aligned}
d_{\ell_i} &= ||\boldsymbol{v}_{\ell_i} - \boldsymbol{v}_{\ell_1}||_2, \\
\theta_{\ell_i} &= \arccos\left((\boldsymbol{v}_{\ell_i} - \boldsymbol{v}_{\ell_1}) \cdot \boldsymbol{z}/d_{\ell_i}\right), \\
\phi_{\ell_i} &= \text{atan2}\left((\boldsymbol{v}_{\ell_i} - \boldsymbol{v}_{\ell_1}) \cdot \boldsymbol{y}, (\boldsymbol{v}_{\ell_i} - \boldsymbol{v}_{\ell_1}) \cdot \boldsymbol{x}\right),
\end{aligned}
\tag{16}
$$

we can derive the transformed spherical coordinate as

$$
\begin{aligned}
d_{\ell_i}' &= ||R\boldsymbol{v}_{\ell_i} + \boldsymbol{t} - (R\boldsymbol{v}_{\ell_1} + \boldsymbol{t})||_2 = ||R(\boldsymbol{v}_{\ell_i} - \boldsymbol{v}_{\ell_1})||_2 = ||\boldsymbol{v}_{\ell_i} - \boldsymbol{v}_{\ell_1}||_2 = d_{\ell_i}, \\
\theta_{\ell_i}' &= \arccos\left((R\boldsymbol{v}_{\ell_i} + \boldsymbol{t} - (R\boldsymbol{v}_{\ell_1} + \boldsymbol{t})) \cdot \boldsymbol{z}'/d_{\ell_i}'\right) = \arccos\left(R(\boldsymbol{v}_{\ell_i} - \boldsymbol{v}_{\ell_1}) \cdot R\boldsymbol{z}/d_{\ell_i}\right) = \theta_{\ell_i}, \\
\phi_{\ell_i}' &= \text{atan2}\left((R\boldsymbol{v}_{\ell_i} + \boldsymbol{t} - (R\boldsymbol{v}_{\ell_1} + \boldsymbol{t})) \cdot \boldsymbol{y}', (R\boldsymbol{v}_{\ell_i} + \boldsymbol{t} - (R\boldsymbol{v}_{\ell_1} + \boldsymbol{t})) \cdot \boldsymbol{x}'\right), \\
&= \text{atan2}\left(R(\boldsymbol{v}_{\ell_i} - \boldsymbol{v}_{\ell_1}) \cdot R\boldsymbol{y}, R(\boldsymbol{v}_{\ell_i} - \boldsymbol{v}_{\ell_1}) \cdot R\boldsymbol{x}\right), \\
&= \text{atan2}\left((\boldsymbol{v}_{\ell_i} - \boldsymbol{v}_{\ell_1}) \cdot \boldsymbol{y}, (\boldsymbol{v}_{\ell_i} - \boldsymbol{v}_{\ell_1}) \cdot \boldsymbol{x}\right), \\
&= \phi_{\ell_i}.
\end{aligned}
\tag{17}
$$

Thus, we show that $f(V) = f(\tau(V))$. Conversely, given spherical coordinates $(d_{\ell_i}, \theta_{\ell_i}, \phi_{\ell_i})$ for fragment center $\ell_i$, we can obtain local Cartesian coordinates through the inverse function $f^{-1}(\cdot)$ as

$$
[d_{\ell_i} \sin(\theta_{\ell_i}) \cos(\phi_{\ell_i}), d_{\ell_i} \sin(\theta_{\ell_i}) \sin(\phi_{\ell_i}), d_{\ell_i} \cos(\theta_{\ell_i})].
\tag{18}
$$

Then, we can reconstruct the original coordinates of fragment centers by applying the $SE(3)$ transformation $(R_{\mathbb{m} \to \mathbb{w}}, \boldsymbol{t}_{\mathbb{m} \to \mathbb{w}})$ between the molecule local frame and the world frame, which is derived in Section 3.3.2. Formally,

$$
\boldsymbol{v}_{\ell_i} = R_{\mathbb{m} \to \mathbb{w}} f^{-1}(\boldsymbol{S})^T + \boldsymbol{t}_{\mathbb{m} \to \mathbb{w}}.
\tag{19}
$$

So, there exist a transformation $\tau \in SE(3)$, such that $f^{-1}(\boldsymbol{S}) = \tau(V)$.

Next, we prove that the rotation vector $\boldsymbol{m}$ between the molecule local frame and the fragment local frame is also $SE(3)$-invariant. Suppose we have the transformation $(R_{\mathbb{m} \to \mathbb{w}}, \boldsymbol{t}_{\mathbb{m} \to \mathbb{w}})$ between the molecule local frame $\mathbb{m}$ and the world frame $\mathbb{w}$, and the transformation $(R_{\mathfrak{g} \to \mathbb{w}}, \boldsymbol{t}_{\mathfrak{g} \to \mathbb{w}})$ between the fragment local frame $\mathfrak{g}$ and the world frame. In Section 3.3.2, we derive the transformation $(R_{\mathfrak{g} \to \mathbb{m}}, \boldsymbol{t}_{\mathfrak{g} \to \mathbb{m}})$ between the fragment local frame and the molecule local frame, such that

$$
R_{\mathfrak{g} \to \mathbb{m}} = R_{\mathbb{m} \to \mathbb{w}}^T R_{\mathfrak{g} \to \mathbb{w}}, \quad \boldsymbol{t}_{\mathfrak{g} \to \mathbb{m}} = R_{\mathbb{m} \to \mathbb{w}}^T (\boldsymbol{t}_{\mathfrak{g} \to \mathbb{w}} - \boldsymbol{t}_{\mathbb{m} \to \mathbb{w}}).
\tag{20}
$$

Given any $SE(3)$ transformation $\tau$ containing rotation matrix $R$ and translation vector $\boldsymbol{t}$, we have

$$
\begin{aligned}
R_{\mathfrak{g} \to \mathbb{m}} &= (R R_{\mathbb{m} \to \mathbb{w}})^T (R R_{\mathfrak{g} \to \mathbb{w}}), \tag{21} \\
&= R_{\mathbb{m} \to \mathbb{w}}^T R^T R R_{\mathfrak{g} \to \mathbb{w}}, \tag{22} \\
&= R_{\mathbb{m} \to \mathbb{w}}^T I R_{\mathfrak{g} \to \mathbb{w}}, \tag{23} \\
&= R_{\mathfrak{g} \to \mathbb{m}}. \tag{24}
\end{aligned}
$$

So we have $R_{\mathfrak{g}\to\mathfrak{m}} = \tau(R_{\mathfrak{g}\to\mathfrak{m}})$. Since $\boldsymbol{m} = g(\mathfrak{m}, \mathfrak{g})$ is derived from $R_{\mathfrak{g}\to\mathfrak{m}}$, as described in Section 3.3.3, we also have $g(\mathfrak{m}, \mathfrak{g}) = g(\tau(\mathfrak{m}, \mathfrak{g}))$.

Conversely, given a rotation vector $\boldsymbol{m}$, we can convert it back to a rotation matrix. Specifically, let $\boldsymbol{m} = \theta\boldsymbol{a}$, where $\theta$ is the rotation angle and $\boldsymbol{a} = (a_x, a_y, a_z)$ is the unit rotation axis. Then, we can construct the skew-symmetric matrix $K$ from $\boldsymbol{a}$ such that

$$K = \begin{bmatrix} 0 & -a_z & a_y \\ a_z & 0 & -a_x \\ -a_y & a_x & 0 \end{bmatrix}. \tag{25}$$

Then, we apply Rodrigues' rotation formula to obtain rotation matrix $R_{\mathfrak{g}\to\mathfrak{m}}$ such that

$$R_{\mathfrak{g}\to\mathfrak{m}} = I + \sin(\theta)K + (1 - \cos(\theta))K^2. \tag{26}$$

Then, we have $R_{\mathfrak{g}\to\mathfrak{w}} = R_{\mathfrak{m}\to\mathfrak{w}}R_{\mathfrak{g}\to\mathfrak{m}}$. From Equation (7), we can obtain $\boldsymbol{t}_{\mathfrak{g}\to\mathfrak{m}}$ and then we can have $\boldsymbol{t}_{\mathfrak{g}\to\mathfrak{w}} = R_{\mathfrak{m}\to\mathfrak{w}}\boldsymbol{t}_{\mathfrak{g}\to\mathfrak{m}} + \boldsymbol{t}_{\mathfrak{m}\to\mathfrak{w}}$. So, we show that there exist a $SE(3)$ transformation $\tau = (R_{\mathfrak{m}\to\mathfrak{w}}, \boldsymbol{t}_{\mathfrak{m}\to\mathfrak{w}})$, such that $g^{-1}(\boldsymbol{m}) = \tau(\mathfrak{m}, \mathfrak{g})$.

$\square$

### C.3 Proof of Theorem 3.4

First we establish a lemma and provide its proof.

**Lemma C.1.** *Let $\mathcal{M}_1 = (V_1, Z_1)$ and $\mathcal{M}_2 = (V_2, Z_2)$ be two 3D molecular graphs, where $z_i$ is the node type vector and $v_i$ is the node coordinates of the molecule $\mathcal{M}_i$. There exists a bijection $b : ver(\mathcal{M}_1) \to ver(\mathcal{M}_2)$ such that for every atom in $\mathcal{M}_1$ indexed $i$, $z_i^{\mathcal{M}_1} = z_{b(i)}^{\mathcal{M}_2}$. Let $\boldsymbol{L}(\mathcal{M})$ be the canonical order of molecule $\mathcal{M}$. For the fragments $\{\mathcal{G}_i\}_{i=1}^k$ of $\mathcal{M}$, we construct canonical order $\boldsymbol{L}(\mathcal{M})$ for the $k$ fragments, $\ell_1, \cdots, \ell_k$, specifically by the order of a fragment's first-ranked atom. Let $\ell_i$ and $\ell_i'$ denote the indexes of the fragment labeled $i$ correspondingly in $\boldsymbol{L}(\mathcal{M}_1)$ and $\boldsymbol{L}(\mathcal{M}_2)$, respectively. Let $f \circ g$ be a surjective function that maps a 3D molecule $\mathcal{M}$ to its spherical representation and rotation vector $(\boldsymbol{S}, \boldsymbol{m})$ under the equivariant molecule local frame, and between molecule local frame and fragment local frame, respectively, following Lemma 3.3. Then the following equivalence holds:*

$$for \ i = 1, \cdots, k, f \circ g(\mathcal{M}_1)_{\ell_i} = f \circ g(\mathcal{M}_2)_{\ell_i'} \iff \mathcal{M}_1 \cong_{3D} \mathcal{M}_2$$

*where $\mathcal{M}_1 \cong_{3D} \mathcal{M}_2$ denotes that $\mathcal{M}_1$ and $\mathcal{M}_2$ are 3D isomorphic.*

*Proof.* From Def. 3.1, we know

$$\mathcal{M}_1 \cong_{3D} \mathcal{M}_2 \iff \begin{cases} \text{there exists bijection } b : ver(\mathcal{M}_1) \to ver(\mathcal{M}_2) \ s.t. \forall i, \ z_i^{\mathcal{M}_1} = z_{b(i)}^{\mathcal{M}_2}, \text{ and} \\ \text{there exists 3D transformation } \tau \in SE(3) \text{ such that } \boldsymbol{v}_{\ell_i'}^{\mathcal{M}_2} = \tau(\boldsymbol{v}_{\ell_i}^{\mathcal{M}_1}), \end{cases}$$

for all $i$. Note that since the fragmentation process is unique, deterministic, and linear, for fragments $\{\mathcal{G}_i\}_{i=1}^k$ of $\mathcal{M}$, $\{\mathcal{G}_i\}_{i=1}^k \equiv \mathcal{M}$. Thus fragmentation does not affect any properties. Thus it's sufficient to prove the equivalence of the two statements on right side with

$$for \ i = 1, \cdots, k, f \circ g(\mathcal{M}_1)_{\ell_i} = f \circ g(\mathcal{M}_2)_{\ell_i'}.$$

Here, the first statement on the right holds by definition. For the second condition, we establish $\tau(\boldsymbol{v}_{\ell_i}) = R\boldsymbol{v}_{\ell_i} + \boldsymbol{t}$. Here $R$ is a rotation matrix, and $\boldsymbol{t}$ is a translation vector.

Following the proof of Lemma 3.3 in Appendix C.2, for any 3D transformation $\tau \in SE(3)$, we have $f(V) = f(\tau(V))$ and $g(\mathfrak{m}, \mathfrak{g}) = g(\tau(\mathfrak{m}, \mathfrak{g}))$. Thus if there exists 3D transformation $\tau \in SE(3)$ such that $\boldsymbol{v}_{\ell_i'}^{\mathcal{M}_2} = \tau(\boldsymbol{v}_{\ell_i}^{\mathcal{M}_1})$, there exists 3D transformation $\tau \in SE(3)$ such that $\boldsymbol{v}_{\ell_i'}^{\mathcal{G}, \mathcal{M}_2} = \tau(\boldsymbol{v}_{\ell_i}^{\mathcal{G}, \mathcal{M}_1})$ for $i = 1, \cdots, k$, and then

$$f \circ g(\mathcal{M}_1)_{\ell_i} = (\boldsymbol{S}_1, \boldsymbol{m}_1) = (\boldsymbol{S}_2, \boldsymbol{m}_2) = f \circ g(\mathcal{M}_2)_{\ell_i'}$$

for $i = 1, \cdots, k$. This proves one direction of the equivalence.

Also following the proof of Lemma 3.3 in Appendix C.2, given spherical representations $\boldsymbol{S} = f(V)$ and rotation vectors $\boldsymbol{m} = g(\mathfrak{m}, \mathfrak{g})$, there exist a transformation $\tau \in SE(3)$, such that $f^{-1}(\boldsymbol{S}) = \tau(V)$ and $g^{-1}(\boldsymbol{m}) = \tau(\mathfrak{m}, \mathfrak{g})$. For $i = 1, \cdots, k$, if we have $f \circ g(\mathcal{M}_1)_{\ell_i} = (\boldsymbol{S}_1, \boldsymbol{m}_1) = (\boldsymbol{S}_2, \boldsymbol{m}_2) = f \circ g(\mathcal{M}_2)_{\ell'_i}$, there exist a transformation $\tau \in SE(3)$, such that $f^{-1}(\boldsymbol{S}_2) = \tau(\boldsymbol{v}_2)$ and $g^{-1}(\boldsymbol{m}) = \tau(\mathfrak{m}_2, \mathfrak{g}_2)$. Similarly, $f^{-1}(\boldsymbol{S}_1) = \tau'(\boldsymbol{v}_1)$ for $\tau' \in SE(3)$, and thus $\tau'(\boldsymbol{v}_1) = f^{-1}(\boldsymbol{S}_1) = f^{-1}(\boldsymbol{S}_2) = \tau(\boldsymbol{v}_2)$. Since $SE(3)$ is a Lie group satisfying the axioms that the set is closed under the binary operation, and that for every element in $SE(3)$, there is an identity inverse. Let the inverse identity of $\tau$ be $\tau^{-1}$. Then $\tau^{-1}\tau(\boldsymbol{v}_2) = \tau^{-1}\tau'(\boldsymbol{v}_1)$, and thus $\boldsymbol{v}_2 = \tau''(\boldsymbol{v}_1)$ with $\tau'' = \tau^{-1}\tau' \in SE(3)$ for $i = 1, \cdots, k$. This proves the other direction of the equivalence.

Therefore, we show that $\mathcal{M}_1 \cong_{3D} \mathcal{M}_2 \iff \forall i, f \circ g(\mathcal{M}_1)_{\ell_i} = f \circ g(\mathcal{M}_2)_{\ell'_i}$ holds. □

Then we prove Theorem 3.4.

**Theorem** (Bijective mapping)**.** *Following Def. 3.1, let $\mathcal{M}_1 = (V_1, Z_1)$ and $\mathcal{M}_2 = (V_2, Z_2)$ be two 3D molecular graphs. Let $\boldsymbol{L}(\mathcal{M})$ be the canonical order for $\mathcal{M}$ and $f(\cdot)$ and $g(\cdot,\cdot)$ be the functions following Lemma 3.3. Let the fragment-position vector for a molecule fragment $\mathcal{G}_i$ be $\boldsymbol{x}_i^* = [s_i, d_i, \theta_i, \phi_i, m_{xi}, m_{yi}, m_{zi}]$, where the vector elements are derived from $f(\cdot)$ and $g(\cdot,\cdot)$ as predefined. For the fragments $\{\mathcal{G}_i\}_{i=1}^k$ of $\mathcal{M}$, we construct canonical order $\boldsymbol{L}(\mathcal{M})$ for the k fragments, $\ell_1, \cdots, \ell_k$, specifically by the order of a fragment's first-ranked atom. We define Frag2Seq : $\mathfrak{M} \to \mathcal{U}$, which maps a molecule $\mathcal{M} \in \mathfrak{M}$ to its sequence representation $U \in \mathcal{U}$, the set of all possible sequence representations, as*

$$\textit{Frag2Seq}(\mathcal{M}) = \textit{concat}(\boldsymbol{x}_{\ell_1}^*, \cdots, \boldsymbol{x}_{\ell_k}^*),$$

*where concat$(\cdot)$ concatenates elements as a sequence. Then Frag2Seq$(\cdot)$ is a surjective function, and the following equivalence holds:*

$$\textit{Frag2Seq}(\mathcal{M}_1) = \textit{Frag2Seq}(\mathcal{M}_2) \Leftrightarrow \mathcal{M}_1 \cong_{3D} \mathcal{M}_2,$$

*where $\mathcal{M}_1 \cong_{3D} \mathcal{M}_2$ denotes $\mathcal{M}_1$ and $\mathcal{M}_2$ are 3D isomorphic. If we allow rounding up spherical coordinate and rotation vector values to $\geq b$ decimal places, then the surjectivity and equivalence still hold, only molecules $\mathcal{M}_1$ and $\mathcal{M}_2$ are $(|10^{-b}|/2)$-constrained 3D isomorphic.*

*Proof.* First, we prove that Frag2Seq : $\mathfrak{M} \to \mathcal{U}$ is a surjective function. Given the definition

$$\text{Frag2Seq}(\mathcal{M}) = \text{concat}(\boldsymbol{x}_{\ell_1}^*, \cdots, \boldsymbol{x}_{\ell_k}^*),$$

where $\boldsymbol{x}_i^* = [s_i, d_i, \theta_i, \phi_i, m_{xi}, m_{yi}, m_{zi}]$, we need to prove that all operations are deterministic. concat$(\cdot)$ and $s_i$ are defined to be deterministic, and following Lemma C.1, $d_i, \theta_i, \phi_i, m_{xi}, m_{yi}, m_{zi}$ are derived from $f \circ g$, which is a function. $\boldsymbol{L}(\mathcal{M})$ outputs $\mathcal{M}$'s canonical order, which is deterministic from the canonical SMILES algorithm. Therefore, Frag2Seq : $\mathfrak{M} \to \mathcal{U}$ is a well-defined function; given a 3D molecule, we can uniquely construct a 1D sequence from Frag2Seq.

Next we prove Frag2Seq's surjectivity. Given any output sequence $q \in \mathcal{U}$ of Frag2Seq, the sequence is in the format

$$q = \textit{concat}([s_1, d_1, \theta_1, \phi_1, m_{x1}, m_{y1}, m_{z1}], ..., [s_k, d_k, \theta_k, \phi_k, m_{xk}, m_{yk}, m_{zk}]).$$

For each fragment in $q$ and its vector $[d_i, \theta_i, \phi_i, m_{xi}, m_{yi}, m_{zi}]$, we follow Lemma 3.3's derivations. Given the surjectivity of the spherical representation function $f$, rotation vector function $g$, and the defined $f^{-1}, g^{-1}$, there must be a unique $\mathcal{M} = (V, Z) \in \mathfrak{M}$ where

$$[d_1, \theta_1, \phi_1, m_{x1}, m_{y1}, m_{z1}], ..., [d_k, \theta_k, \phi_k, m_{xk}, m_{yk}, m_{zk}] = f \circ g(\mathcal{M}).$$

Therefore, $\forall$ output sequence $q \in \mathcal{U}$ there exists

$$\mathcal{M} = (V, Z) \in \mathfrak{M} \quad \textit{s.t.} \quad q = \text{Frag2Seq}(\mathcal{M}),$$

*i.e.*, Frag2Seq is surjective; given a sequence output of Frag2Seq, we can uniquely reconstruct a 3D molecule.

Since all the above proof does not involve numerical comparisons and 3D isomorphic, they also hold if we allow rounding up spherical coordinate and rotation vector values to $\geq b$ decimal places.

Now we prove the equivalence $\text{Frag2Seq}(\mathcal{M}_1) = \text{Frag2Seq}(\mathcal{M}_2) \Leftrightarrow \mathcal{M}_1 \cong_{3D} \mathcal{M}_2$, starting from right to left. We also cover the constrained case where we allow rounding up values to $\geq b$ decimal places. When a number is truncated after $b$ decimal places, according to the rounding principle, the maximum error caused is $\epsilon \leq \frac{|10^{-b}|}{2}$. If $\mathcal{M}_1 \cong_{3D} \mathcal{M}_2$ (or $\mathcal{M}_1 \cong_{3D-\frac{|10^{-b}|}{2}} \mathcal{M}_2$), *i.e.*, molecules $\mathcal{M}_1$ and $\mathcal{M}_2$ are 3D isomorphic (or $(|10^{-b}|/2)$-constrained 3D isomorphic), then from Lemma 3.2 we know the canonical forms $\boldsymbol{L}(\mathcal{M}_1) = \boldsymbol{L}(\mathcal{M}_2)$. For the fragments $\{\mathcal{G}_i\}_{i=1}^k$ of $\mathcal{M}$, we construct canonical order $\boldsymbol{L}(\mathcal{M})$ for the $k$ fragments, $\ell_1, \cdots, \ell_k$, specifically by the order of a fragment's first-ranked atom. Let $\ell_i$ and $\ell_i'$ denote the indexes of the fragment labeled $i$ correspondingly in $\boldsymbol{L}(\mathcal{M}_1)$ and $\boldsymbol{L}(\mathcal{M}_2)$, respectively. Since molecules $\mathcal{M}_1$ and $\mathcal{M}_2$ are 3D isomorphic (or $(|10^{-b}|/2)$-constrained 3D isomorphic), from Def.3.1 we know $\forall i$ of $\{\mathcal{G}_i\}_{i=1}^k$, $s_{\ell_i} = s_{\ell_i'}$; and from Lemma C.1 we know for $i = 1, \cdots, k$, $f \circ g(\mathcal{M}_1)_{\ell_i} = f \circ g(\mathcal{M}_2)_{\ell_i'}$ (or satisfies with $\frac{|10^{-b}|}{2}$ error range allowed for each numerical value). Thus, we have

$$
\begin{aligned}
&\text{Frag2Seq}(\mathcal{M}_1) \\
&= \text{concat}(\boldsymbol{x}_{\ell_1}^*, \cdots, \boldsymbol{x}_{\ell_k}^*) \\
&= \text{concat}_{s_i(\mathcal{G}_i), \mathcal{G}_i \in \mathcal{M}_1, d_i, \theta_i, \phi_i, m_{xi}, m_{yi}, m_{zi} \in f \circ g(\mathcal{M}_1), i=1,\ldots k}([s_{\ell_i}, d_{\ell_i}, \theta_{\ell_i}, \phi_{\ell_i}, m_{x\ell_i}, m_{y\ell_i}, m_{z\ell_i}]) \\
&= \text{concat}_{s_i(\mathcal{G}_i), \mathcal{G}_i \in \mathcal{M}_2, d_i, \theta_i, \phi_i, m_{xi}, m_{yi}, m_{zi} \in f \circ g(\mathcal{M}_2), i=1,\ldots k}([s_{\ell_i'}, d_{\ell_i'}, \theta_{\ell_i'}, \phi_{\ell_i'}, m_{x\ell_i'}, m_{y\ell_i'}, m_{z\ell_i'}]) \\
&= \text{concat}(\boldsymbol{x}_{\ell_1'}^*, \cdots, \boldsymbol{x}_{\ell_k'}^*) \\
&= \text{Frag2Seq}(\mathcal{M}_2),
\end{aligned}
$$

$$(27)$$

where $s_i(\cdot)$ is the canonical SMILES string. Therefore, we have shown that if two molecules are 3D isomorphic (or within the round-up error range $\frac{|10^{-b}|}{2}$), their sequences resulting from Frag2Seq must be identical.

Finally, we prove the equivalence from left to right. We provide proof by contradiction. Given that $\text{Frag2Seq}(\mathcal{M}_1) = \text{Frag2Seq}(\mathcal{M}_2)$, we assume that the molecules $\mathcal{M}_1$ and $\mathcal{M}_2$ are not 3D isomorphic (or not $\frac{|10^{-b}|}{2}$-constrained 3D isomorphic). We denote with $\mathcal{M}_1 = (V_1, Z_1)$ and $\mathcal{M}_2 = (V_2, Z_2)$. If $\mathcal{M}_1$ and $\mathcal{M}_2$ are not even isomorphic for $Z_i$, then from Def.3.1, there does not exist a node-to-node mapping from $\mathcal{M}_1$ to $\mathcal{M}_2$, where each atom is identical in atom number. Then from Lemma 3.2, we know the canonical forms $\boldsymbol{L}(\mathcal{M}_1) \neq \boldsymbol{L}(\mathcal{M}_2)$. Thus for $\text{Frag2Seq}(\mathcal{M}_1) =$

$$\text{concat}_{s_i(\mathcal{G}_i), \mathcal{G}_i \in \mathcal{M}_1, d_i, \theta_i, \phi_i, m_{xi}, m_{yi}, m_{zi} \in f \circ g(\mathcal{M}_1), i=1,\ldots k}([s_{\ell_i}, d_{\ell_i}, \theta_{\ell_i}, \phi_{\ell_i}, m_{x\ell_i}, m_{y\ell_i}, m_{z\ell_i}]),$$

and $\text{Frag2Seq}(\mathcal{M}_2) =$

$$\text{concat}_{s_i(\mathcal{G}_i), \mathcal{G}_i \in \mathcal{M}_2, d_i, \theta_i, \phi_i, m_{xi}, m_{yi}, m_{zi} \in f \circ g(\mathcal{M}_2), i=1,\ldots k}([s_{\ell_i'}, d_{\ell_i'}, \theta_{\ell_i'}, \phi_{\ell_i'}, m_{x\ell_i'}, m_{y\ell_i'}, m_{z\ell_i'}]),$$

there must be at least one pair of $s_{\ell_i}, s_{\ell_i'}$ where $s_{\ell_i} \neq s_{\ell_i'}$. Therefore, $\text{Frag2Seq}(\mathcal{M}_1) \neq \text{Frag2Seq}(\mathcal{M}_2)$, which is a contradiction to the initial condition and ends the proof.

If $\mathcal{M}_1$ and $\mathcal{M}_2$ are isomorphic for $Z_i$, we continue with the following analyses. Let $\ell_i$ and $\ell_i'$ denote the indexes of the fragment labeled $i$ correspondingly in $\boldsymbol{L}(\mathcal{M}_1)$ and $\boldsymbol{L}(\mathcal{M}_2)$, respectively. Let $f \circ g$ be the surjective function mapping a 3D graph to its spherical and rotation representations. Since $\mathcal{M}_1$ and $\mathcal{M}_2$ are not 3D isomorphic (or not even $\frac{|10^{-b}|}{2}$-constrained 3D isomorphic), from Lemma C.1, we know there exists at least one

$$i \in \{1, \cdots, k\}, s.t. f \circ g(\mathcal{M}_1)_{\ell_i} \neq f(\mathcal{M}_2)_{\ell_i'},$$

(or even with error range $\frac{|10^{-b}|}{2}$ allowed), otherwise, we would have

$$\forall i \in \{1, \cdots, k\}, f(\mathcal{M}_1)_{\ell_i} = f(\mathcal{M}_2)_{\ell_i'} \Rightarrow \mathcal{M}_1 \cong_{3D} \mathcal{M}_2,$$

contradicting the above condition. Thus for $\text{Frag2Seq}(\mathcal{M}_1) =$

$$\text{concat}_{s_i(\mathcal{G}_i), \mathcal{G}_i \in \mathcal{M}_1, d_i, \theta_i, \phi_i, m_{xi}, m_{yi}, m_{zi} \in f \circ g(\mathcal{M}_1), i=1,\ldots k}([s_{\ell_i}, d_{\ell_i}, \theta_{\ell_i}, \phi_{\ell_i}, m_{x\ell_i}, m_{y\ell_i}, m_{z\ell_i}]),$$

and $\text{Frag2Seq}(\mathcal{M}_2) =$

$$\text{concat}_{s_i(\mathcal{G}_i), \mathcal{G}_i \in \mathcal{M}_2, d_i, \theta_i, \phi_i, m_{xi}, m_{yi}, m_{zi} \in f \circ g(\mathcal{M}_2), i=1,\ldots k}([s_{\ell_i'}, d_{\ell_i'}, \theta_{\ell_i'}, \phi_{\ell_i'}, m_{x\ell_i'}, m_{y\ell_i'}, m_{z\ell_i'}]),$$

at least one pair of spherical coordinates does not correspond, so there must be at least one pair of $(d_{\ell_i}, \theta_{\ell_i}, \phi_{\ell_i}, m_{x\ell_i}, m_{y\ell_i}, m_{z\ell_i})$ and $(d_{\ell'_i}, \theta_{\ell'_i}, \phi_{\ell'_i}, m_{x\ell'_i}, m_{y\ell'_i}, m_{z\ell'_i})$ where

$$(d_{\ell_i}, \theta_{\ell_i}, \phi_{\ell_i}, m_{x\ell_i}, m_{y\ell_i}, m_{z\ell_i}) \neq (d_{\ell'_i}, \theta_{\ell'_i}, \phi_{\ell'_i}, m_{x\ell'_i}, m_{y\ell'_i}, m_{z\ell'_i})$$

(or

$$\min(|d_{\ell_i}, \theta_{\ell_i}, \phi_{\ell_i}, m_{x\ell_i}, m_{y\ell_i}, m_{z\ell_i}| - |d_{\ell'_i}, \theta_{\ell'_i}, \phi_{\ell'_i}, m_{x\ell'_i}, m_{y\ell'_i}, m_{z\ell'_i}|) > \frac{|10^{-b}|}{2}).$$

Thus, $\text{Frag2Seq}(\mathcal{M}_1) \neq \text{Frag2Seq}(\mathcal{M}_2)$, which contradicts the initial condition. Therefore, we have shown that if two constructed sequences from Frag2Seq are identical, their corresponding molecules must be 3D isomorphic. This ends the proof.

$\square$

