# OpenReview forum: "Fragment and Geometry Aware Tokenization of Molecules for Structure-Based Drug Design Using Language Models"
_ICLR.cc/2025/Conference — ICLR 2025 Poster_

### Official Review · Reviewer_AJ93 · 2024-10-31

**Soundness:** 2
**Presentation:** 3
**Contribution:** 2
**Rating:** 6
**Confidence:** 4

**Summary:**

This paper introduces Frag2Seq, a fragment and geometry-aware language model designed for structure-based drug design. The model employs a SE(3)-equivariant approach to decompose 3D molecular structures into fragments and represents these fragments as sequences while preserving their geometric information. Protein embeddings are extracted using the pre-trained inverse folding model ESM-IF1. To effectively capture protein-ligand interactions, Frag2Seq incorporates cross-attention mechanisms between the molecule embeddings and the protein embeddings within the language model. Experiments on the CrossDocked dataset show that Frag2Seq outperforms the baselines on  metrics such as Vina Score, QED, and Lipinski The method also exhibits superior sampling efficiency, with up to approximately 300 times faster generation compared to atom-based autoregressive and diffusion baselines.

**Strengths:**

1. The method proposed in the paper significantly outperforms recent state-of-the-art methods in terms of drug-likeness and high binding affinity of the generated molecules, demonstrating its superiority in generating drug candidates.
2. The sample efficiency of Frag2Seq is markedly higher than that of existing methods, offering substantial speedup in the generation process. This efficiency is crucial for accelerating the drug discovery pipeline and reducing computational costs.
3. The paper innovatively leverages the established ordering with canonical SMILES for 3D-isomorphic molecules, which is of paramount importance for sequence models. This approach ensures that the fragments of molecules  are accurately represented and compared in a sequence format.

**Weaknesses:**

1. The use of SE(3)-invariant frames to encode the 3D structure of molecules, while effective, is a common approach in  protein structure design and thus does not introduce a significant innovation in this paper. The methodology primarily leverages the canonical ordering of SMILES strings to concatenate fragment sequences, which is a straightforward application. Similarly, the use of cross-attention to directly integrate protein and small molecule information is also quite direct. As a result, this work is more incremental in nature rather than presenting a breakthrough in methodology.
2. In terms of input processing, the quantization of continuous structural information into discrete tokens results in a loss of precision. This could potentially affect the model's ability to capture fine-grained details of molecular structures. On the output side, the language model is unable to directly generate information about chemical bonds, which necessitates the use of additional models to infer this critical information. This dependency on secondary models for bond inference introduces an extra step in the generation process and may impact the overall coherence and accuracy of the generated molecules.

**Questions:**

1. Since the language model does not directly generate chemical bond information, how reliable is the subsequent model used for bond inference, and how is its performance validated? Do baseline models also require bond inference, and if so, do they use the same model? Please elaborate more on the rationality behind the experimental design to ensure that the performance of drug design is accurately assessed.
2. Could you explain why a language model was chosen over other methods, such as 3D atom-based diffusion models, for the task of structure-based drug design? What are the theoretical and practical benefits?
3. Is the direct use of cross-attention between the entire protein and molecule sequences a simplistic approach, or can experiments demonstrate that it effectively captures key information about the binding pocket?

---

### Official Review · Reviewer_bkQt · 2024-11-02

**Soundness:** 3
**Presentation:** 3
**Contribution:** 2
**Rating:** 6
**Confidence:** 3

**Summary:**

The authors propose a fragment-based bijective mapping between 3D molecules and canonical SE(3)-invariant sequences. They do so by combining molecule and fragment local frames and representing fragments as spherical coordinates w.r.t to the molecule frame. They prove that the representation preserves structural completeness and geometric invariance. They adopt the representation for LM-based conditional generation of ligands for protein pockets, demonstrating competitive results with a wide set of baselines.

**Strengths:**

- The proposed sequence representation respects structural completeness and geometric invariance, and the authors provide rigorous proofs for the claims.
- The SBDD baselines are comprehensive, with the method achieving superior results in efficacy and efficiency
- The authors include sufficient ablations and analysis for the design choices of their method.

**Weaknesses:**

- The approach is not entirely novel (though its application to SBDD is)
- It's unclear why this text-based representation and LM-based generation is needed for SBDD
- Alternate strategies for incorporating protein context should be compared with

**Questions:**

- In lines 248-249, the authors write "...with the key to be the canonical SMILES of each fragment and the value to be the atom types and atom local coordinates in each fragment under the associated fragment local frame." => why is the key only the canonical SMILES? Would different conformations of the same fragment be represented by the same key?

---

### Official Review · Reviewer_DBhm · 2024-11-03

**Soundness:** 2
**Presentation:** 2
**Contribution:** 2
**Rating:** 3
**Confidence:** 5

**Summary:**

This paper proposes Frag2Seq, which leverages fragment and geometric information to enhance structure-based drug design. By transforming 3D molecules into fragment sequences and using SE(3)-equivariant frames to preserve the geometry of fragments, this model aims to improve the efficiency of molecule generation. The paper also incorporates protein pocket embeddings using cross-attention to capture protein-ligand interactions and demonstrates some performance improvements.

**Strengths:**

1. Innovation in fragment-based generation: The model’s use of fragment-based generation reduces the computational complexity of molecule generation.
2. Incorporation of protein embeddings: The use of protein pocket embeddings to enhance the targeting of drug molecule generation demonstrates a promising attempt at capturing protein-ligand interactions.
3. Generation efficiency: The method improves generation speed over traditional atom-level generation approaches.

**Weaknesses:**

1. Lack of innovation: The model structure closely resembles that of IPDiff, with the only significant change being the replacement of the diffusion mechanism with a language model, limiting the novelty of the approach.
2. Insufficient binding affinity accuracy: The model underperforms in binding affinity compared to IPDiff, which achieves higher Vina scores.
3. Lack of critical comparative analysis: There are no direct accuracy comparisons with state-of-the-art models, like IPDiff, preventing a full evaluation of the model's advantages.
4. Limited practicality and scalability: Although fragment generation improves computational efficiency, it is limited in producing high-accuracy molecules and binding affinity, which reduces its practicality.

**Questions:**

1. Could the authors provide more detailed experimental results, especially in terms of binding affinity and molecular quality, to further validate the model’s effectiveness?
2. How does this model perform on CrossDocked2020, in comparison with IPDiff?
3. Is it possible to consider integrating an interaction prior mechanism similar to IPDiff within the diffusion model to enhance generation accuracy and practical applicability?

---

### Official Review · Reviewer_3w4Z · 2024-11-08

**Soundness:** 3
**Presentation:** 3
**Contribution:** 3
**Rating:** 8
**Confidence:** 3

**Summary:**

The paper presents Frag2Seq, a method for structure-based drug design using language models (LMs) to generate drug-like molecules based on 3D fragments. By transforming 3D molecules into SE(3)-invariant sequences which can used with LMs, Frag2Seq captures essential geometric information. Additionally, it incorporates protein context through cross-attention with pre-trained protein embeddings, enabling effective target-aware molecule generation. Experiments show that Frag2Seq shows competitive performance against baselines in binding affinity, drug-likeness (QED), and synthesis accessibility, with sampling speeds up to 300 times faster than atom-based generative models.

**Strengths:**

1. Frag2Seq effectively incorporates protein context through cross-attention with pre-trained protein embeddings, allowing it to generate target-specific molecules. This approach addresses a common limitation in SBDD by aligning molecular generation more closely with real-world drug discovery where small molecule and protein interactions play a vital role in binding.
2. The SE(3)-invariant tokenization method preserves important 3D geometric information, offering a promising way to leverage the generative capabilities of LLMs. This geometric awareness could enhance accuracy in predicting drug-like properties, such as binding affinity and drug-likeness.
3.  Frag2Seq demonstrates strong performance on metrics like binding affinity and drug-likeness while achieving up to 300x speedup over atom-based approaches.

**Weaknesses:**

1. For the models to be practically usable in applications like drug discovery, the scoring mechanism for generated molecules needs greater clarity on how specific molecular characteristics (e.g., fragment size, functional groups) influence metrics like vina scores. In other words the paper can focus a bit more on model interpretability.
2. The fragment-based generation approach, which splits molecules by rotatable bonds while preserving rigid functional groups (e.g., hydroxy, carboxyl), may limit flexibility in positioning critical binding groups, potentially affecting target-specific binding. How robust are the fragmentation rules in Section 3.2?

**Questions:**

1. How robust and adaptable are the fragmentation rules in Section 3.2 when applied to structurally complex molecules? Could this fragmentation style lead to a loss of important molecular interactions?
2. The authors should consider providing an attention map visualization highlighting the interactions between protein pocket features and small molecule. This could  help enhance interpretability, and improve practical usability in drug discovery applications

---

### Comment · Reviewer_bkQt · 2025-02-08

congrats on your acceptance!

---

### Meta-Review · Area_Chair_2dr2 · 2024-12-21

**Metareview:**

This paper proposes a method to generate molecules in SE(3) equivariant way using SE(3) invariant tokenization of molecular fragments. The tokenization is based on canonicalized ordering of atoms (and fragments) and serialization based on fragments and relative rotations between the fragment-wise local frames.

The paper is promising as a way to use language models for 3D generation of molecules.

From my perspective, the proposed idea is not entirely novel, since the idea to decompose a molecule into 3D fragments and learn a generative model to choose the 3D fragments and assemble them has been considered before. The main novelty lies in using a language model instead of diffusion model for the generation. The current presentation of the paper is not crystal clear about this connection, which could be improved. The authors should also provide more thorough empirical evaluation clearly explaining why and how the language model perform better than the diffusion models.

Despite the weakness, I recommend acceptance for this paper. The authors provide convincing argument that the proposed idea is useful. While I am not entirely convinced that the proposed method is SOTA, the work may lead to future follow-ups that use autoregressive generation for 3D molecules, which I believe to be an underexplored area.

**Additional Comments On Reviewer Discussion:**

Reviewer DBhm was the only negative reviewer with valid points about the weakness of the paper. However, while I resonate with the concerns, I do not think the weakness is enough to reject the paper. This also aligns with the assessment of Reviewer bkQt.

---

### Decision · Program_Chairs · 2025-01-22

Accept (Poster)